# How to Train Your LLM Web Agent:
# A Statistical Diagnosis

**Dheeraj Vattikonda**[*,1,2,5]   **Santhoshi Ravichandran**[*,1,2]   **Emiliano Penaloza**[*,1,2,6]

Hadi Nekoei[1,2,6]   Megh Thakkar[1]   Thibault Le Sellier de Chezelles[1,2,3]   Nicolas Gontier[1]

Miguel Muñoz-Mármol[1]   Sahar Omidi Shayegan[1,2,5]   Stefania Raimondo[1]

Xue Liu[2,5]   Alexandre Drouin[1]   Laurent Charlin[2,4]   Alexandre Piché[1]

Alexandre Lacoste[1]   **Massimo Caccia**[*,1]

## Abstract

LLM-based web agents have recently made significant progress, but much of it has occurred in closed-source systems, widening the gap with open-source alternatives. Progress has been held back by two key challenges, first, a narrow focus on single-step tasks that overlooks the complexity of multi-step web interactions, and second, the high compute costs required to post-train LLM-based web agents. To address this, we present the first statistically grounded study on compute allocation for LLM web-agent post-training. Our approach uses a two-stage pipeline, training a Llama 3.1 8B student to imitate a Llama 3.3 70B teacher via SFT, followed by on-policy reinforcement learning. We find this process highly sensitive to hyperparameter choices in setting where exhaustive sweeps are impractical. To spare others from expensive trial-and-error, we sample 1,370 configurations and use bootstrapping to estimate effective hyperparameters. Our results show that combining SFT with on-policy RL consistently outperforms either approach alone on both WorkArena and MiniWob++. Further, this strategy only requires 55% of the compute to match the peak of pure SFT on MiniWob++, pushing the compute–performance Pareto frontier and is the only strategy that can close the gap with closed-source models.

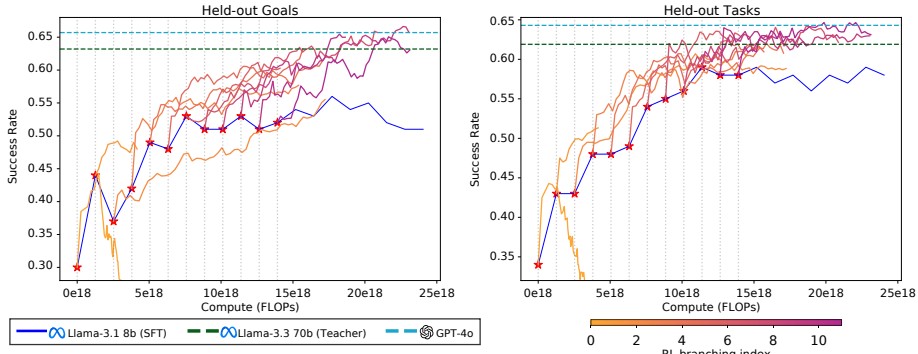

Figure 1: **Compute–performance frontier on MiniWoB++ (results averaged over two seeds).** The blue curve shows pure SFT on teacher demonstrations. Warm-colored curves represent hybrid runs that branch off from SFT checkpoints and continue training with RL. Early transitions to RL push the Pareto frontier achieving higher success rates for the same compute and is the only approach able to achieve over 30% improvement on **both** held-out goals (left) and held-out tasks (right) closing the gap between open and closed-source models. See fig. 4 for the corresponding plot with Qwen2.5 7B.

[*]Equal contribution   [1]ServiceNow Research   [2]Mila–Quebec AI Institute   [3]Polytechnique Montréal   [4]HEC Montréal   [5]McGill University   [6]Univeristé de Montréal

39th Conference on Neural Information Processing Systems (NeurIPS 2025).

# 1 Introduction

Large language model (LLM) agents for web interfaces have advanced rapidly, but open-source systems still trail proprietary ones. Bridging this gap would allow organizations to train smaller, cost-efficient agents tailored to their needs while maintaining data privacy. Yet, despite impressive progress of open-source models in domains like math and code generation, advances in training web-capable LLM agents remain limited by mainly by the lack of attention to multi-turn, long-horizon tasks and the high cost and low reproducibility of current training pipelines.

Most research centers on single-step tasks like code or math domains with rapid feedback and simplified credit assignment which fall short of real-world web environments requiring sequential decisions and long-horizon planning. Recent benchmarks like WebArena [29], WorkArena [10], OSWorld [26], and The Agent Company [27] have exposed how brittle current methods become under delayed rewards, sparse feedback, and compounding errors. Addressing these settings demands not just better agents, but reproducible, compute-efficient training pipelines an area we directly tackle.

However, building such pipelines is nontrivial. Modern LLM post-training often involves a combination of Supervised Fine-Tuning (SFT) and Reinforcement Learning (RL), with performance sensitive to a large number of interacting hyperparameters [13]. Single-seed results are noisy and misleading, but exhaustive tuning and multi-seed sweeps (e.g. [2]) remain infeasible for most labs due to the cost of LLM training. This makes it all the more critical to design pipelines that are not only effective but statistically robust and accessible within realistic compute budgets.

In this paper, we tackle these gaps by providing a statistically driven diagnosis of training LLM agents for web-based, multi-step tasks. Specifically, we study how to allocate compute between expensive, high-quality off-policy demonstrations from a large teacher model and cheaper on-policy rollouts from a smaller student model. We analyze this tradeoff across two levels of generalization, *held-out goals* being tasks encountered during training but with novel goals and *held-out tasks*, which are entirely unseen during training.

To study this compute-performance trade-off we use a two-stage training pipeline. First, a LLaMA 3.3 70B teacher model generates $K$ successful trajectories to warm-start a smaller LLaMA 3.1 8B student via SFT. We then branch of various SFT checkpoints where training continues with an on-policy RL phase using Group Relative Policy Optimization (GRPO) [9]. Our central objective is to determine the *optimal compute allocation and hyperparameter* mix for training web-based LLM agents. To this end, we run 1,370 training configurations, varying key hyperparameters and branching points between SFT and RL. We then apply bootstrap-based analysis to estimate the impact of different hyper-parameters on downstream performance and how they vary across branching SFT checkpoints. This data-driven approach enables us to identify important considerations to get the most out of each run. We use this method to show the optimal mix between SFT and RL in the MiniWob++ environment, achieving better task accuracy at a significantly lower cost. Additionally, we provide concrete recommendations on compute allocations between SFT and RL on the more demanding WorkArena environment.

Putting things together, we show how our study yields several actionable insights. First, branching into RL early, but not immediately after SFT leads to better outcomes. This hybrid strategy consistently outperforms pure SFT and pure RL, and reaches the maximal performance of pure SFT while requiring only 55% of the compute, effectively pushing the compute–performance Pareto frontier. It is also the only strategy that can close the gap with closed-source models. Second, curriculum learning and error log feedback help the less SFT warmup has been applied but can become counterproductive thereafter. Third, in GRPO, applying zero-advantage filtering consistently improve performance, while dividing the advantage with the standard deviation and wether to use an importance ratio are dependent on the amount of SFT warmup done. Fourth, decoding temperature is consistently critical, while learning rate and discount rate must also be carefully tuned.

These findings are significant for two reasons. First, they give smaller research groups a reproducible, budget-aware playbook for pushing open LLM agents closer to state-of-the-art without scaling model size. Second, they address a slice of the broader reproducibility concerns recently highlighted in the RL community [3, 13], offering a template for rigorous, statistically-grounded hyperparameter tuning and compute allocation in LLM agent training.

## 2 Background

This section consolidates the algorithmic ingredients used throughout the paper: (i) the MDP formulation of web-based language agents, (ii) SFT on expert traces, (iii) GRPO for RL, and (iv) curriculum and normalization techniques that stabilize training.

### 2.1 Language Agents as MDPs

We model each task as a Markov Decision Process (MDP) $\mathcal{M} = \langle \mathcal{S}, \mathcal{A}, P, r, \rho_0, \gamma \rangle$. A state $\mathbf{s}_t \in \mathcal{S}$ is a textual context–in our case a prompt and an action $a_t \in \mathcal{A}$ is a textual response generated by the agent. Each action $a_t$ consists of a sequence of tokens $o_{1_t:K_t}$ sampled autoregressively from the policy $\pi_\theta(a_t|\mathbf{s}_t)$ parameterized by an LLM sampled with temperature $\rho_{\mathrm{LLM}}$. The environment then returns a scalar reward $r_t \in \{-1, 1\}$ indicating task failure/success. In our setting we assume the environment dynamics are $p(\mathbf{s}_{t+1} \mid a_t) \in P$. We optimize the policy $\pi_\theta$ to maximize the expected discounted return:

$$J(\theta) = \mathbb{E}_{\tau \sim \pi_\theta} \left[ \sum_{t=0}^{T} \gamma^t r_t \right] \tag{1}$$

Here, $\gamma \in [0, 1]$ is the discount rate, which controls the agent's preference for immediate versus future rewards and $\tau = (s_0, a_0, r_0, s_1, a_1, r_1, \ldots, s_T)$ refers to a trajectory sampled from the policy.

### 2.2 Off-policy boot-strapping via SFT

We first imitate a stronger *expert* policy $\pi_E$ by minimizing the cross-entropy loss

$$\mathcal{L}_{\mathrm{SFT}}(\theta) = -\mathbb{E}_{\tau \sim \pi_E} \left[ \sum_{t=0}^{T} \log \pi_\theta(a_t \mid s_t) \right]. \tag{2}$$

SFT offers a high-quality, low-variance gradient but is inherently *off-policy* which can lead to poor generalization [6].

### 2.3 Rejection Fine-Tuning

Similar to SFT, Rejection Fine-Tuning (RFT) optimizes the policy using successful trajectories, but in this case bootstrapped of the training policy rather than a larger expert

$$\mathcal{L}_{\mathrm{RFT}}(\theta) = -\mathbb{E}_{\tau \sim \pi_\theta} \left[ \mathbf{1}_{\{r_t=1\}} \sum_{t=0}^{T} \log \pi_\theta(a_t \mid s_t) \right]. \tag{3}$$

### 2.4 On-Policy Improvement with Multi-Turn GRPO

After the initial SFT warmup phase, we continue training using on-policy RL using GRPO as the optimization algorithm. Like REINFORCE [21], GRPO maximizes the expected return using a policy gradient objective. However, GRPO introduces additional structure by leveraging per-goal advantage normalization and importance weighting, wherein each trajectory is associated with a specific goal in our context, referring to the seed of the target task. For a given goal $g$, the group-normalized advantage function is:

$$A_{t,g} = \frac{r_{t,g} - \mathrm{mean}(\mathbf{R}_{t,g})}{\mathrm{std}(\mathbf{R}_{t,g})},$$

where $\mathbf{R}_{t,g} = (r_{t,1}, \ldots, r_{t,G})$ is the set of rewards across the $G$ goals at timestamp $t$. In addition to this similar to proximal policy optimization (PPO) [2] an importance-ratio is often applied to the GRPO objective:

$$\eta_{t,g} = \frac{\pi_\theta(a_t \mid s_t)}{\pi_{\theta_{\mathrm{old}}}(a_t \mid s_t)},$$

where $\pi_\theta$ is the current policy and $\pi_{\theta_{\text{old}}}$ is the behavior policy used to collect trajectories. Finally, a clipped-minimum is also applied to stabilize the training process. Putting things together the GRPO objective for our multi-turn setting is:

$$\mathcal{J}_{\text{MT-GRPO}}(\theta) = \mathbb{E}_{\tau \sim \pi_{\theta_{\text{old}}}} \left[ \frac{1}{G} \sum_{g=1}^{G} \frac{1}{T} \sum_{t=0}^{T} \min\left( \eta_{t,g} A_{t,g}, \text{clip}(\eta_{t,g}, 1-\epsilon, 1+\epsilon) A_{t,g} \right) \right]. \tag{4}$$

Traditionally the GRPO objective includes a KL penalty between the optimizing and reference policies which we do not use as early experiments showed they did not improve performance, slowed down training and required additional compute budget.

**Zero-advantage filtering.** Tokens with $A_{t,g} = 0$ contribute no learning signal yet still consume memory. Dropping them yields a constant *effective* batch size and modestly accelerates training [28].

### 2.5 Curriculum through Variance-Aware Boltzmann Sampling

To promote steady learning progress, we design a curriculum that prioritizes challenging tasks, neither trivial nor too difficult [22]. Specifically, we select tasks according to a Boltzmann distribution centered around a target return $\mu_{\text{target}}$ which specifies the desired performance threshold, encouraging focus on partially mastered tasks, with a temperature parameter $\rho_{\text{Curr}}$ controlling the sharpness of the distribution, with lower values concentrating probability mass tightly around $\mu_{\text{target}}$.

This sampling mechanism dynamically adapts the training distribution, concentrating learning on tasks where the agent is neither already proficient nor entirely unskilled. As a result, the agent avoids premature convergence on easy tasks and prevents wasted effort on tasks far beyond its current capabilities.

## 3 Methodology

Our training pipeline consists of two sequential stages SFT followed by RL framed as a resource allocation problem. We also describe our hyperparameter sweep strategy and statistical analysis protocol that consolidates hundreds of runs into reliable conclusions. We evaluate our recipe along two axes: compute cost, measured in FLOPs (using the formula from [1]), and model performance, assessed on both *unseen* training goals and held-out testing tasks.

**Stage 1 – Supervised Fine-Tuning (SFT).** We begin by generating $N_E$ expert trajectories using a large teacher model. Only successful trajectories are retained after filtering, and the corresponding $(s, a)$ pairs, along with chain-of-thought annotations, form the SFT dataset. Note that computing the cost of the SFT dataset includes both successful and discarded unsuccessful trajectories.

We then train a smaller student model for $T_{\text{SFT}}$ gradient steps. To explore the trade-off between SFT and RL, we branch off $B$ times at fixed intervals along the SFT trajectory, yielding checkpoints at timesteps $t_b \in [0, T_{\text{SFT}}]$. Each checkpoint corresponds to a student policy $\pi_{\theta(t_b)}$, which initializes a distinct RL phase. The total compute used up to each branching point, $F_{\text{SFT}}(t_b)$, includes both teacher inference and student training FLOPs accumulated over $t_b$ steps.

**Stage 2 – RL Fine-Tuning.** Each policy $\pi_{\theta(t_b)}$ is further trained using GRPO for $T_{\text{RL}}$ steps. The compute cost of this phase, $F_{\text{RL}}(T_{\text{RL}})$, includes both the FLOPs required for data collection (online rollouts) and for student updates across all $T_{\text{RL}}$ steps. The total FLOPs for a full training run starting from $\theta(t_b)$ is computed as:

$$\text{FLOPs}(t_b) = F_{\text{SFT}}(t_b) + F_{\text{RL}}(T_{\text{RL}}). \tag{5}$$

By varying $t_b$ throughout SFT, we assess how shifting compute between expert-driven supervision and on-policy learning impacts final performance. This setup highlights the trade-off between the high compute cost of expert supervision and the lower-cost, but noisier, nature of on-policy learning.

### 3.1 Estimating the Uncertainty of the Hyperparameter Selection Process

Across the different SFT checkpoints, we sample 1,370 distinct configurations with ten varying hyperparameters (see Appendix L for details). Our objective is to study the effect of various

**Algorithm 1** Bootstrap Estimation of Hyperparameter Importance

**Require:** Set of training runs `Results`, evaluation metric column $M$, hyperparameter of interest $H$, number of bootstrap iterations $B$

1: $\mathcal{W} = \texttt{dict}(\ )$ ▷ Win counts per HP value
2: $\mathcal{S} = \texttt{defaultdict(List)}$ ▷ Score samples per HP value
3: $H_{\text{vals}} = \texttt{unique(Results}[H])$ ▷ Distinct values of the hyperparameter
4: $\texttt{weights} = \{\ h_i : 1/\texttt{count(Results}[h_i])$ for each $h_i \in H_{\text{vals}}\ \}$
5: **for** $b = 1$ to $B$ **do**
6: $\quad \texttt{sample}_b = \texttt{resample(Results, weights)}$
7: $\quad h^* = \text{argmax}_h\ \texttt{sample}_b[M]$ ▷ Get the winning HP value for $\texttt{sample}_b$
8: $\quad \mathcal{W}[h^*] += 1$
9: $\quad$ **for** each $h$ in $H_{\text{vals}}$ **do**
10: $\quad\quad \mathcal{S}[h]+ = \max \texttt{sample}_b[\texttt{sample}_b[\mathcal{H}] == h][M]$
11: $\quad$ **end for**
12: **end for**
13: **for** each $h$ in $H_{\text{vals}}$ **do**
14: $\quad$ Compute 95% confidence interval from $\mathcal{S}[h]$
15: **end for**
16: **return** $\mathcal{W}$, $\mathcal{S}$ with confidence intervals

hyperparameters (HP) on the downstream success rate of our trained agents. This comes with two important considerations. First, if we change the value of, e.g., the batch size, and we want to know if a bigger batch size is better, the learning rate and other parameters need to be readjusted close to their optimal configuration (under a fixed budget). Secondly, to account for noise, we would need to restart the same experiment several times to avoid spurious conclusions. In practice, this is out of reach. For a more computationally friendly approach, we resort to bootstrapping the collection of trials over different HP configurations. The boostrap algorithm has many desriable properties such as being a consistent and unbiased estimator for the sampling distribution of any data statistic without requiring parametric assumptions (see Appendix K for details), thus allowing us to reliably obtain robust estimates of both the expected value and variance of the success rate given by any HP.

**Bootstrapping the hyperparameter selection process.** From the full set of 1,370 training runs, we perform bootstrap resampling by drawing individual runs (with replacement). For each resample, we identify the best-performing configuration and repeat this process 1,000 times. We also compute the fraction of times each hyperparameter value "wins", which serves as an estimated probability that it belongs to the global optimum. This procedure serves two purposes: to estimate the *maximum* relative improvement a specific hyper-parameter provides while accounting for variation in other parameters and to offer uncertainty estimates in the form of win-rate distributions between different configurations. In addition, to better understand how optimal hyper-parameters may change depending on the amount of SFT warmup we apply this analysis across various SFT checkpoints.

**Balancing unequal coverage.** Due to random search, some HP values were explored more than others, biasing the winner toward the larger groups. To correct for this, each run is sampled with probability $\propto 1/$group size, approximating an equal compute budget for every HP value.

We provide a detailed explanation of this procedure in Algorithm 1. Additionally in Appendix J we describe how use this procedure to obtain the results reported in Figure 1.

## 4 Experimental Setup

In this section, we describe the experimental setup used to validate our findings. We detail the models, benchmarks, action spaces, training framework, compute infrastructure, and evaluation protocols, ensuring that all components are aligned with the goals of studying compute-efficient, reproducible training for LLM web agents.

**Models.** We evaluate our approach using two teacher models, Llama 3.3 70B and Qwen 2.5 72B to generate SFT traces, with Llama 3.1 8B and Qwen 2.5 7B acting as the student model for fine-tuning.

All models operate with a 16k token context window to handle the complexity of web-based tasks. For the LLama models we report results using GRPO, while for Qwen we report results using RFT as we found it more stable during training.

**Benchmarks.** Our experiments focus on two benchmarks. The first is MiniWoB++, a suite of 30 medium-horizon web interaction tasks, where we observe that optimal policies typically complete tasks in 2 to 5 steps. The second is WorkArena [10], a more challenging benchmark of 33 enterprise knowledge-work tasks, where we observe optimal policies generally complete tasks in 3 to 10 steps. These benchmarks provide a representative spectrum of sequential decision-making challenges faced by interactive LLM agents. Both benchmarks are depicted in Figure 7.

**Observation & Action Spaces.** MiniWoB++ provides raw HTML trees, whereas WorkArena leverages accessibility trees (AxTrees), which we truncate to 16k tokens to meet hardware constraints. The agent operates in a discrete action space composed of high-level UI primitives: `noop`, `fill(node, text)`, `click(node)`, `type(node, text)`, `select_option(node, option)`, `scroll(node)`, and `hover(node)`. This abstraction allows the agent to interact effectively with diverse web interfaces. All agents employ chain-of-thought prompting [25]. We also experiment with applying *error log feedback*, allowing the agent to receive explicit error messages when it takes invalid actions (during both training and testing).

**Training Framework.** To manage the training pipeline, we use BROWSERGYM [8] for orchestrating Chromium-based web environments and structuring the agent's action space, while AGENTLAB [8] handles agent design. Model fine-tuning is conducted with TORCH-TUNE [23], utilizing Fully Sharded Data Parallelism (FSDP) to enable scalable training across multiple GPUs. Given the high memory demands of long-sequence processing, we apply activation offloading and gradient checkpointing techniques, achieving approximately 40% reduction in memory usage.

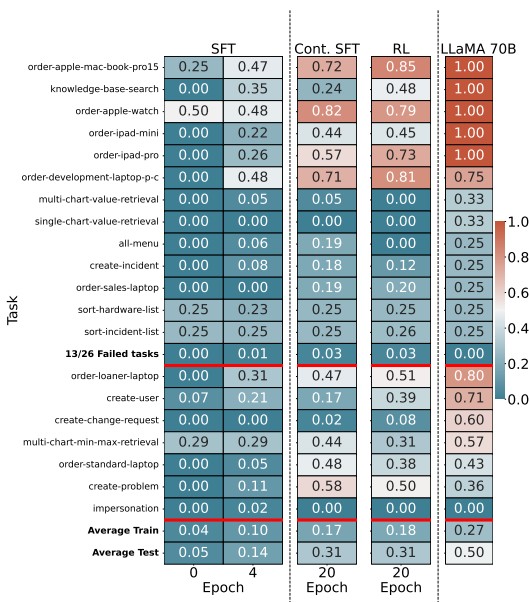

Figure 2: Per-task performance of SFT and SFT+RL agents on WorkArena. The Llama 3.1 8B model is initially fine-tuned for 4 epochs on trajectories from a teacher Llama 3.3 70B model. Training then continues either with additional SFT or with GRPO fine-tuning up to epoch 20. The teacher model's success rate is also shown.

**Evaluation Protocol.** Both environments are stochastic, and we control for randomness during training by fixing the task seed. Each task–seed pair is referred to as a "goal". The evaluation protocol assesses generalization at two levels: (i) performance on held-out goals within the training tasks, and (ii) performance on entirely new, unseen held-out tasks. We consider both settings important as they capture different aspects of generalization held-out goals test *reliability* on trained tasks, while held-out tasks test the model's ability to *transfer* skills learnt in training to entirely new situations.

In both settings we use the the average task success rate as the evaluation metric, which reflects the agent's ability to generalize beyond its training distribution. To reduce the impact of evaluation noise during model selection, we apply a rolling average with a window size of 3 on the held-out goals, and select the checkpoint with the highest smoothed score.

For all our reported runs (SFT, RL, SFT + RL) we report the average of the non-smoothed scores at the selected checkpoint, aggregated over four runs chosen through the described model selection procedure based on our random search. For all other models, we report average scores over 100 independent runs.

Table 1: Comparison of our method with baselines on the held-out train and test splits of **MiniWoB++** and **WorkArena**. Here $\pm$ are the averaged standard errors computed separately for each of the two top seeds.

| Model | MiniWoB++ | | WorkArena | |
|---|---|---|---|---|
| | Held-out Goals | Held-out Tasks | Held-out Goals | Held-out Tasks |
| Claude-3.5-Sonnet | **70.5**$_{\pm 2.0}$ | **70.4**$_{\pm 4.3}$ | 52.5$_{\pm 3.2}$ | **70.0**$_{\pm 5.5}$ |
| GPT-4o | 65.7$_{\pm 2.1}$ | 64.3$_{\pm 4.5}$ | 42.1$_{\pm 3.2}$ | 55.7$_{\pm 5.9}$ |
| GPT-4o-Mini | 56.2$_{\pm 2.2}$ | 66.1$_{\pm 4.4}$ | 27.1$_{\pm 2.9}$ | 28.6$_{\pm 5.4}$ |
| Llama-3.1-70B-Instruct | 57.0$_{\pm 2.2}$ | 65.2$_{\pm 4.4}$ | 25.0$_{\pm 2.8}$ | 32.9$_{\pm 5.6}$ |
| o1-Mini | 69.7$_{\pm 2.1}$ | 66.1$_{\pm 4.4}$ | **53.8**$_{\pm 3.2}$ | 68.6$_{\pm 5.5}$ |
| Llama-3.1-405B-Instruct | 65.9$_{\pm 2.4}$ | 65.2$_{\pm 2.4}$ | 39.2$_{\pm 2.4}$ | 58.6$_{\pm 2.5}$ |
| Llama-3.3-70B Instruct *(Teacher)* | 63.2$_{\pm 2.4}$ | 61.9$_{\pm 2.4}$ | 36.0$_{\pm 2.4}$ | 44.0$_{\pm 2.5}$ |
| Llama-3.1-8B Instruct *(Student)* | 29.5$_{\pm 2.3}$ | 36.4$_{\pm 2.4}$ | 8.3$_{\pm 1.5}$ | 4.2$_{\pm 1.2}$ |
| Llama-3.1-8B RL *(Ours)* | 43.5$_{\pm 3.5}$ | 43.5$_{\pm 3.5}$ | 8.0$_{\pm 1.9}$ | 11.5$_{\pm 2.3}$ |
| Llama-3.1-8B SFT *(Ours)* | 55.2$_{\pm 2.5}$ | 56.7$_{\pm 2.5}$ | 28.4$_{\pm 2.0}$ | 26.4$_{\pm 2.0}$ |
| Llama-3.1-8B SFT+RL *(Ours)* | **67.2**$_{\pm 2.1}$ | **63.3**$_{\pm 2.2}$ | **35.4**$_{\pm 2.1}$ | **28.8**$_{\pm 2.0}$ |
| Qwen-2.5-72B Instruct *(Teacher)* | 61.0$_{\pm 3.4}$ | 59.0$_{\pm 3.4}$ | 33.3$_{\pm 2.4}$ | 27.0$_{\pm 2.5}$ |
| Qwen-2.5-7B Instruct *(Student)* | 32.8$_{\pm 3.3}$ | 37.0$_{\pm 3.4}$ | 5.2$_{\pm 2.2}$ | 10.0$_{\pm 3.8}$ |
| Qwen-2.5-7B RL *(Ours)* | 52.5$_{\pm 3.5}$ | 53.0$_{\pm 3.5}$ | 4.5$_{\pm 1.5}$ | 7.0$_{\pm 1.8}$ |
| Qwen-2.5-7B SFT *(Ours)* | 57.0$_{\pm 3.5}$ | 56.5$_{\pm 3.5}$ | 25.0$_{\pm 4.3}$ | 21.0$_{\pm 4.1}$ |
| Qwen-2.5-7B SFT+RL *(Ours)* | **65.0**$_{\pm 3.4}$ | **61.5**$_{\pm 3.4}$ | **32.5**$_{\pm 3.3}$ | **25.0**$_{\pm 3.1}$ |

**Compute Allocation Protocol.**   We track the total floating-point operations (FLOPs) consumed during both SFT and GRPO phases, following the procedure in Section 3 and Appendix D. For RL branching, we start from the SFT run that (i) attains the highest SFT performance and (ii) exhibits stable learning, allowing us to sample checkpoints at regular intervals and cleanly study the effect of SFT warm-up. Because this run sits in the top 10% of all SFT trials by area-under-the-training-curve (AUC) the mean success rate across all checkpoints of the run, serving as a proxy for overall training efficiency, we match that selection pressure for RL by averaging the top two seeds out of ten RL trials, thus approximating the 90th-percentile of the RL distribution and ensuring a fair compute-aware comparison between strategies.

## 5   Main Results and Compute Trade-Offs

In this section, we present our primary findings and analyze the trade-offs between SFT and RL in terms of both performance and compute efficiency.

**Performance Overview.**   Table 1 summarizes the results on MiniWoB++ and WorkArena. Combining SFT with RL consistently yields the best performance among student models. Pure SFT and pure RL each fall short, with RL from scratch particularly struggling due to sparse rewards and unstable learning dynamics. These results highlight the complementary strengths of expert demonstrations and on-policy fine-tuning.

On MiniWoB++, this approach not only maximizes student performance but also matches the teacher and significantly closes the gap with proprietary models. In contrast, WorkArena remains more challenging: while SFT+RL improves over SFT alone, student performance still lags behind the teacher and proprietary models, suggesting that stronger supervision or more effective RL strategies may be needed for complex enterprise tasks.

We observe that agent performance eventually saturates for both SFT and SFT+RL, especially on more difficult tasks. Further analysis of this saturation behavior is provided at the end of this section. Notably, WorkArena's test set can be easier than its training set for certain agents, which adds nuance to the observed performance gaps and complicates straightforward interpretation of generalization performance.

**Compute–Performance Trade-Off.**   Our analysis focuses on the trade-off between costly but high-quality teacher demonstrations and cheaper, noisier on-policy rollouts. To examine this, we branch RL fine-tuning from SFT checkpoints sampled at fixed intervals along the training trajectory.

Figure 1 illustrates the compute–performance frontier on MiniWoB++, where runs that combine SFT with RL (warm colors) consistently outperform pure SFT (blue), pushing the Pareto frontier forward. Early branching into RL unlocks substantial gains in compute efficiency, delivering stronger performance at lower compute budgets. For example, SFT+RL reaches the maximal performance of pure SFT on the test set (achieved at approximately 11 exaFLOPs with pure SFT) using only around 6 exaFLOPs translating to roughly 45% of compute saved (11 vs 6 exaFLOPs).

Notably, this is the only strategy that closes the performance gap with closed-source models like GPT-4o. The trend is consistent across both held-out goals and held-out tasks: warm-started RL yields higher performance than either SFT or RL alone, reinforcing the importance of blending expert supervision with on-policy learning. These findings underscore the need to balance sample efficiency from demonstrations with the compute efficiency of on-policy learning, and the results generalize to Qwen2.5 7B (see fig. 4).

**Task Performance Saturation and Analysis**    Despite extensive post-training, agent performance on the WorkArena benchmark plateaus at around 40% after just 9–10 epochs. This stagnation appears to stem from the intrinsic difficulty of certain tasks (e.g as sorting and filtering tasks) that even frontier models struggles to solve (see Figure 2). A per-task breakdown shows that while both SFT and RL agents gradually close the performance gap with the teacher model, with RL achieving a slightly higher final success rate, a significant portion of tasks (14 out of 33) remain completely unsolved. These failures are attributed to either the limitations of the teacher model or the sparsity of reward signals, both of which hamper the learning process. On-policy RL exploration proves ineffective in overcoming these challenges due to the lack of foundational skills and informative feedback. These findings underscore the need for more effective methods to address complex tasks under sparse reward settings. Additional per-task performance results for WorkArena and Miniwob are provided in Appendix C.

**Shortcut-Seeking Agents and Task Integrity**    During extended training we found that giving agents full administrative privileges invites creative but problematic shortcuts. For instance, in WorkArena's "Order hardware devices" task, rather than navigating through the catalogue as intended, the agent simply edits its homepage to add a direct link to the ordering form. While this hack fulfills the success condition, it violates the task's spirit and, worse, the modifications persist across sessions. Other agents entering the environment later inherit the altered UI, leading to failures unrelated to their own policies.

In general, agents exploited admin rights to create or delete elements, reshaping pages in ways that saved clicks but destabilised the environment. These findings argue for a tighter sandbox enough access for exploration and accomplishment, but safeguards against permanent, instance-wide side-effects.

# 6    Ablation and Sensitivity Analysis

We simulate re-running hyperparameter configurations and selecting the best-performing ones using the method described in Section 3.1. This is done across three checkpoints: the base LLaMA 3.1 8B Instruct model and two warm-started variants with an additional $2.5 \times 10^{18}$ and $7.6 \times 10^{18}$ FLOPs of supervised fine-tuning, respectively, to assess variations across compute budgets. We evaluate the held-out goals and held-out tasks performance of 10 hyper-parameters across 1,370 runs. We report final held-out task performance to verify generalization in Appendix G finding no significant deviations between held-out tasks and held-out goals parameters.

Figure 3 displays our findings, which we summarize as follows. *Curriculum learning* is beneficial when starting RL from scratch but becomes detrimental after warm-starting, likely because warm-started models already perform well on easy tasks, and curriculum forces them to overfocus on harder ones. *Error log feedback* helps when there's no SFT but otherwise does not, likely because SFT warmup removes many common errors made by weaker models. A *decoding temperature* of 0.25 consistently yields the best results, striking a balance between exploration and exploitation; lower values led to under-exploration and were discarded. *Grouped-relative advantage* helps only after SFT, while using raw rewards works better when starting directly from the Instruct model, possibly due to how advantage scaling interacts with the initial weights. *Zero-advantage filtering* improves training across most settings by ensuring batches focus on informative updates. *Standard-deviation*

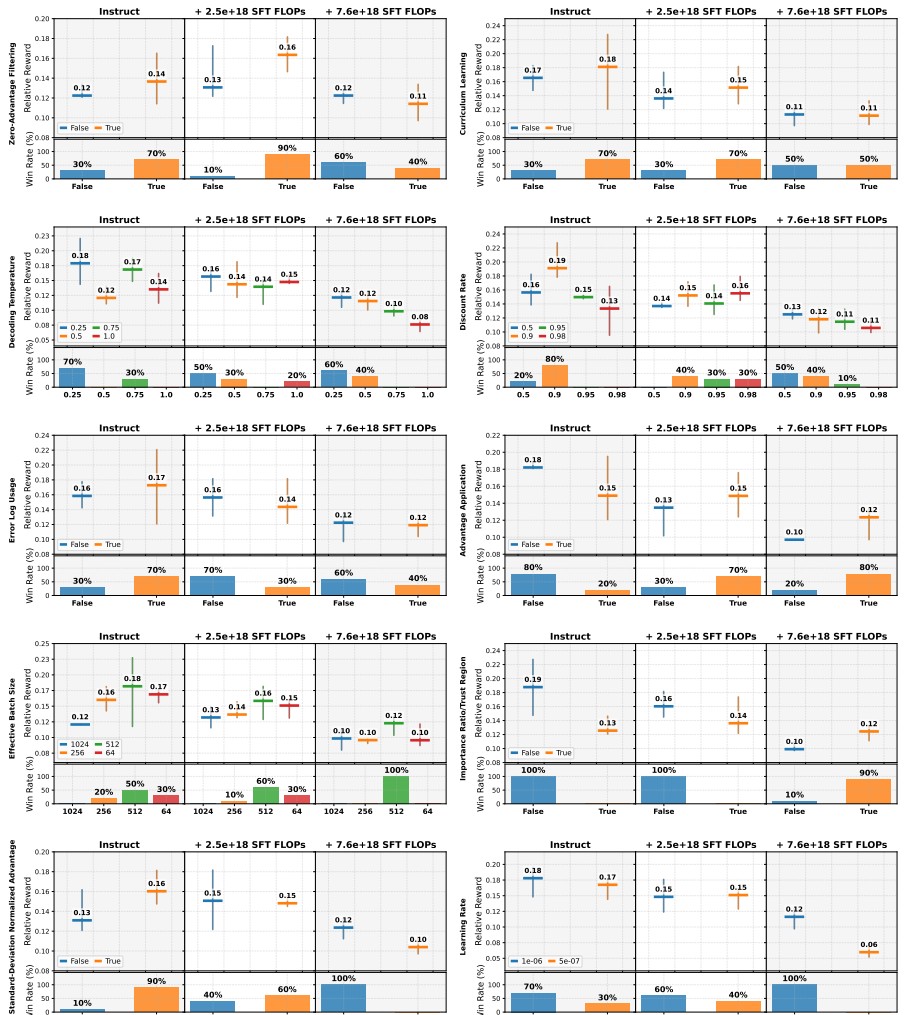

Figure 3: Bootstrap analysis ($n = 1000$ samples) of hyperparameter optimization across different SFT compute budgets on *training* held out tasks. Each subplot examines a different hyperparameter, including increasing SFT compute: the base instruct model (left), $+2.5 \times 10^{18}$ SFT FLOPs (middle), and $+7.6e \times 10^{18}$ SFT FLOPs (right). For each hyperparameter-compute combination, the top panel shows relative reward performance with error bars indicating 95% confidence intervals, while the bottom panel displays win rates representing the percentage of bootstrap iterations where each parameter value achieved maximum performance. Results demonstrate that optimal hyperparameter values shift as model pre-training compute increases, suggesting that hyperparameter selection should be adapted to the computational budget allocated to SFT.

*normalized advantage*, as noted by [17], seems to aid performance under less finetuned models and decrease in value the more finetuning is done. *Importance ratio correction and trust region*, though standard, also hurt models with little or no SFT, likely because conservative updates slow down learning. In contrast, for models that start from a stronger SFT checkpoint, these mechanisms can help stabilize training and avoid catastrophic updates. For the *learning rate*, the larger value (1e-6) generally worked better. *Effective batch size* of 512 appears to be generally be a safe and robust choice in all our experiments. Finally, regarding the *discount rate*, values above 0.9 work well for models with little or no SFT warmup, while heavily warm-started models benefit from a lower rate around 0.5 likely because it encourages the agent to optimize more aggressively on tasks it already handles well. A further hypothesis is that the large deviation in optimal hyperparameters between the base and warm-start models could be attributed to the significant entropy reduction introduced by SFT (see Appendix H).

# 7 Related Work

**Best prectices in deep RL.**   Building on the recognition of reproducibility challenges and unstable RL training of LLM agents, recent studies have proposed best practices for training LLM agents using RL methods. Dang and Ngo [7] recommend leveraging high quality data, balancing easy and hard problems, and controlling length generation with cosine reward. Yu et al. [28] promote higher clipping in the GRPO loss to promote diversity and avoid entropy collapse, dynamic sampling to improve training efficiency and stability, token level gradients for long CoT sequences, and overlong reward shaping to reduce reward noise. Roux et al. [20] introduce tapered variant of importance sampling to speed up learning while maintaining stable learning dynamics. The proposed method (TOPR) allows the handling of both positive and negative examples in a fully offline setting. More generally, Hochlehnert et al. [13] emphasizes the need for greater methodological precision, particularly concerning decoding parameters, random seeds, prompt formatting, as well as the hardware and software frameworks, to guarantee transparent and thorough assessments of model performance. These practices are essential for developing robust and reproducible agents.

**LLM Agents trained with RL on multi-step environments.**   Recent advancements have sought to bridge the gap in training LLM agents for multi-step environments, with approaches like WebRL [19] and SWEET-RL [30] demonstrating significant progress. WebRL employs a self-evolving curriculum to address the challenges of sparse feedback and task scarcity, successfully enhancing the performance of open LLMs in web-based tasks [19]. Similarly, SWEET-RL introduces a hierarchical structure that enables effective credit assignment over multiple turns, improving policy learning and generalization in collaborative reasoning tasks [30]. These studies collectively illustrate the necessity of adapting RL techniques to accommodate the complexities of multi-step interactions, paving the way for more capable and versatile LLM agents. Similar in spirit to our work, [4] propose an empiral study on the inference cost of trained LLM web agents. An extended version of the related work is provided in Appendix F.

**LLM agents trained with RL in multi-step environments.**   Recent work has begun closing the gap for LLM agents that must reason over multiple steps. WebRL [19] introduces a self-evolving curriculum that tackles sparse feedback and task scarcity, substantially boosting open-source agents on web tasks. SWEET-RL [30] adds a hierarchical credit-assignment scheme spanning many turns, improving both learning stability and generalization in collaborative reasoning. Together, these studies underscore the need to adapt RL techniques to the intricacies of long-horizon interaction, paving the way for more capable, versatile agents. Similar in spirit, [4] provide an empirical analysis of inference costs for trained LLM web agents. An extended discussion of related work appears in Appendix F.

# 8 Discussion

**Limitations.**   Our focus is on providing a comprehensive perspective on training an LLM-based web agent, studying compute trade-offs, hyperparameter selection, and analyzing failure cases. With this in mind, our results are limited to English-language web interfaces and Llama 3 models in the 8B–70B parameter range, where larger models may alter trade-offs.

Regarding our statistical method, it does not account for the lack of coverage from the random search. A more exhaustive search could discover configurations that would change the conclusions drawn in this study. We note also that a significant portion of the reported uncertainty is due to epistemic uncertainty that could be reduced by evaluating more configurations.

**Conclusion.**   We present a statistically grounded study on training LLM web agents, analyzing the trade-off between SFT and RL. We perform a random sweep across 1,370 configurations to identify optimal hyperparameter choices, finding interesting variation across compute budgets. Using the bootstrap prescribed hyper-parameters we show that branching into RL early, but not immediately, after SFT achieves better performance–compute trade-offs, matching peak SFT with 45% less compute and closing the gap with closed-source agents. Our findings offer a reproducible, budget-aware blueprint for advancing open-source LLM web agents in complex multi-step environments.

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

## Broader Impact

Web-based LLM agents have the potential to revolutionize markets by enabling more cost-efficient and effective workflows. Our work focuses on making these agents accessible across a range of compute budgets, empowering not only industrial labs but also smaller research groups and individuals to train their own assistants. This approach promotes data privacy and reduces reliance on costly infrastructure.

Despite their promise, web-based agents face significant challenges that limit their broader adoption. Issues such as reliability, vulnerability to adversarial attacks, and limited access to proprietary data remain key obstacles to realizing their full potential.


# A    Compute-peformance frontier for Qwen2.5 7B

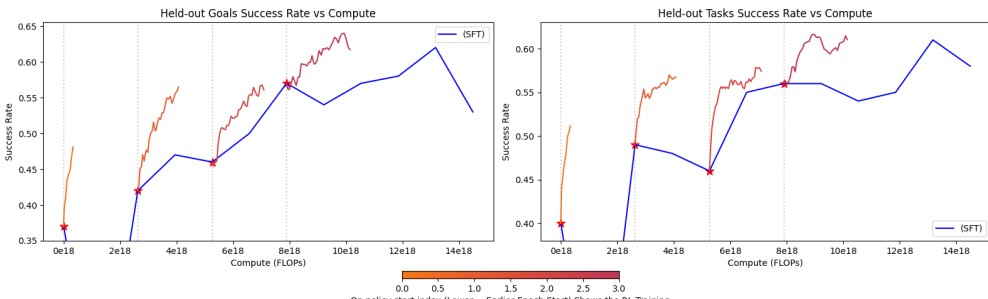

Figure 4: **Compute–performance frontier on MiniWoB++ (results averaged over two seeds) for Qwen2.5 7B.** The blue curve shows pure SFT on teacher demonstrations. Warm-colored curves represent hybrid runs that branch off from SFT checkpoints and continue training with RL. Early transitions to RL push the Pareto frontier achieving higher success rates for the same compute and is the only approach able to achieve over $30\%$ improvement on **both** held-out goals (left) and held-out tasks (right).

# B    Compute Infrastructure

Our computational infrastructure comprises $8 \times$ H100-80GB GPUs for expert data generation with the 70B model. For student model training, we allocate $2 \times$ H100 GPUs for MiniWoB++ experiments and $4 \times$ H100 GPUs for WorkArena experiments, reflecting the increased complexity of the latter.

# C    Extended Learning and Saturation Analysis

**Challenges in Agent-Environment Interaction**    In this section we talk about the general challenges faced by the agent to interact effectively with the environment.

- **Observation/action space mismatch**: One of the important thing to note specifically in our web environment is the observation space which the agent uses is a bit different from the action space. Multiple times, the agent can see the correct action in the AxTree but the action space, the icon is not visible and to make it visible, the agent needs to scroll down and do the action. This mismatch causes huge problems [15]

- **UI Misuse**: The agent tries to interact with items in the environment in ways that it is not designed. For example, the agent trying to fill in a checkbox with value True while it should just click on it. [18]

- **Repeating actions**: A common issue we observed is the repetition of actions across multiple consecutive steps, often accompanied by verbose and redundant chains of thought. The agent frequently restates similar thoughts or re-executes the same actions unnecessarily, leading to inefficiencies and sometimes getting stuck in loops. [18].

# D    Deriving Compute Cost

**FLOPs Estimation Methodology** Flop calculations are based on model architecture, token counts, and average sequence lengths observed during training and evaluation.

**FLOPs per Token**

We estimate FLOPs per token using the following formula, adapted from nvidia benchmarking[1]:

$$\text{FLOPs}_{\text{per token}} = (\text{FLOPs}_{\text{attn}} + \text{FLOPs}_{\text{MLP}} + \text{FLOPs}_{\text{embed}}) \times (1 + \text{backward multiplier}) \quad (6)$$

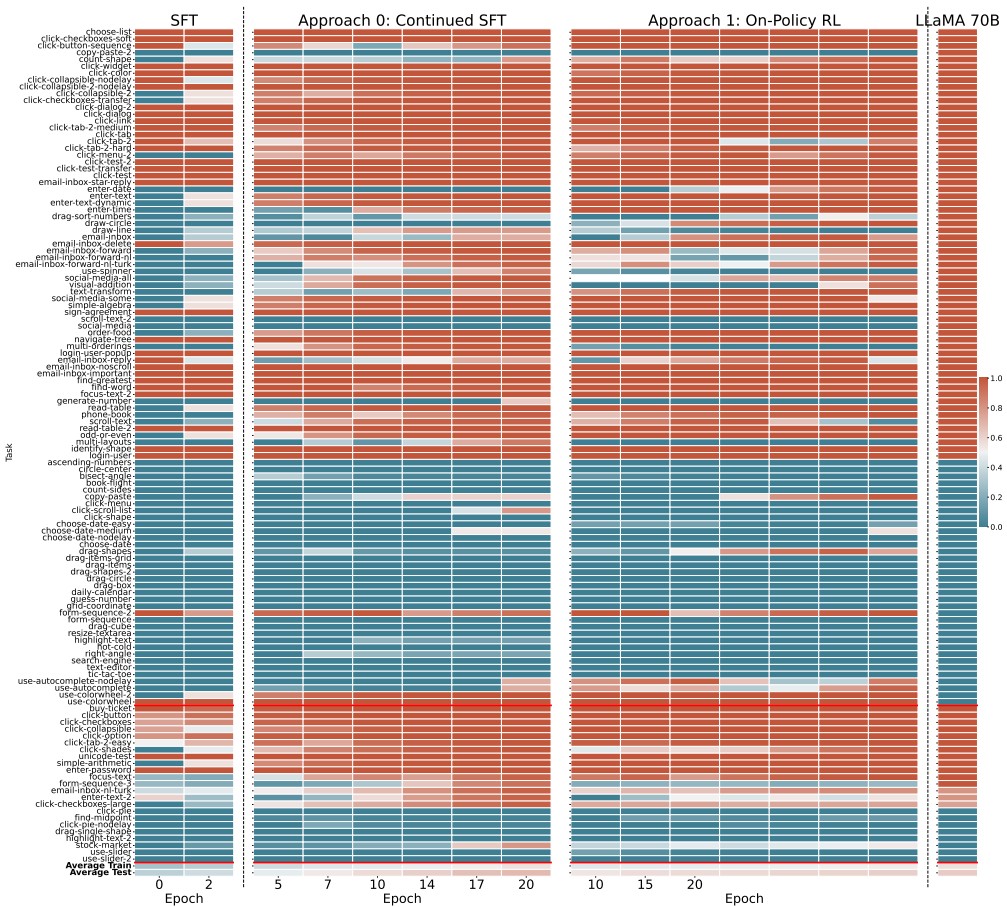

Figure 5: Per task performance of SFT and SFT+RL agents on MiniWob++.

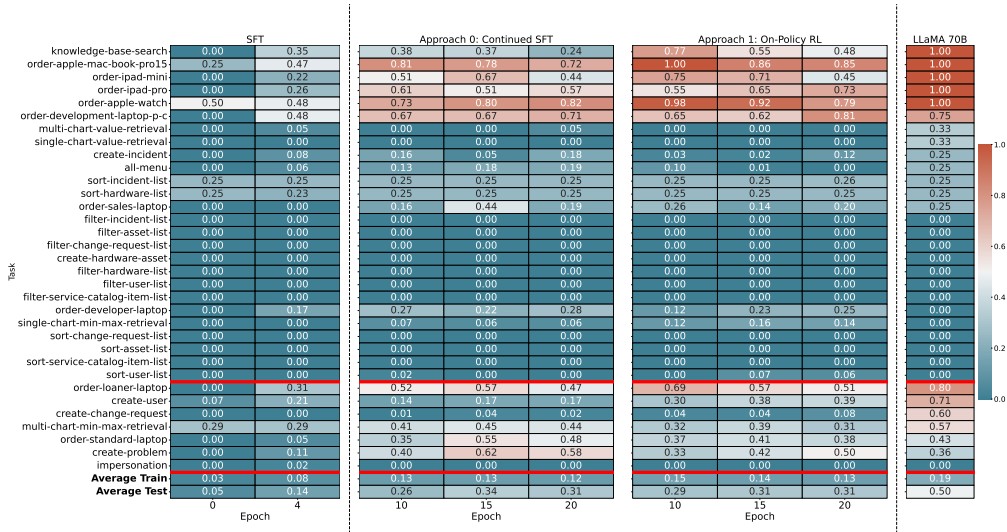

Figure 6: Per task performance of SFT and SFT+RL agents on WorkArena.

Where:

$$\text{FLOPs}_{\text{attn}} = 12 \times (\text{number of layers}) \times (\text{hidden size})^2$$
$$\times \left( 1 + \frac{\text{number of query groups}}{\text{number of attention heads}} + \frac{\text{sequence length}}{\text{hidden size}} \right) \quad (7)$$

$$\text{FLOPs}_{\text{MLP}} = 18 \times (\text{number of layers}) \times (\text{hidden size}) \times \text{FFN} \quad (8)$$
$$\text{FLOPs}_{\text{embed}} = 6 \times \text{vocabulary size} \times (\text{hidden size}) \quad (9)$$

**On-Policy FLOPs (LLaMA-8B)**

We compute the total FLOPs for each on-policy epoch by summing the training and testing FLOPs:

$$\text{FLOPs}_{\text{train}} = N_{\text{train}} \times \text{FLOPs}_{\text{per token}}^{(\text{backward}=3)} \quad (10)$$

$$\text{FLOPs}_{\text{test}} = N_{\text{test}} \times \text{FLOPs}_{\text{per token}}^{(\text{backward}=0)} \quad (11)$$

$$\text{FLOPs}_{\text{epoch}} = \text{FLOPs}_{\text{train}} + \text{FLOPs}_{\text{test}} \quad (12)$$

Where $N_{\text{train}}$ and $N_{\text{test}}$ are the number of tokens used for training and evaluation respectively. Sequence length $S$ is measured per epoch from logged metrics.

**Offline FLOPs (Generation: LLaMA-70B, Training: LLaMA-8B)**

Offline training includes two compute components:

- **Data Generation** (LLaMA-70B, forward-only):

$$\text{FLOPs}_{\text{gen}} = N_{\text{gen}} \times \text{FLOPs}_{\text{per token}}^{(70B,\text{backward}=0)} \quad (13)$$

  where $N_{\text{gen}}$ = avg seq len $\times$ samples per epoch (from dataset metadata).

- **Training** (LLaMA-8B, with backward pass):

$$\text{FLOPs}_{\text{train}} = N_{\text{gen}} \times \text{FLOPs}_{\text{per token}}^{(8B,\text{backward}=3)} \quad (14)$$

The total FLOPs per offline epoch is:

$$\text{FLOPs}_{\text{epoch}} = \text{FLOPs}_{\text{gen}} + \text{FLOPs}_{\text{train}} \quad (15)$$

All FLOPs values are reported in **exaFLOPs** by dividing the total FLOPs by $10^{18}$.

# E    Benchmark Descriptions

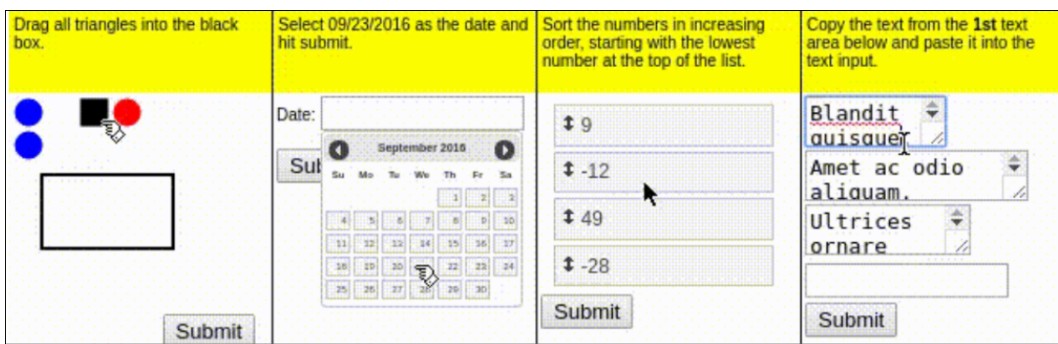

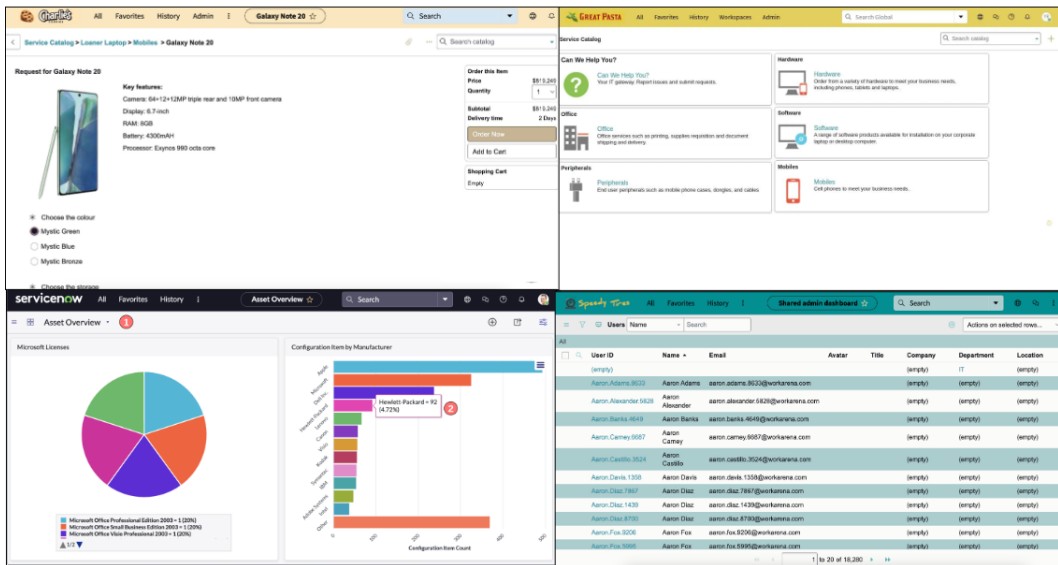

Figure 7: Example tasks in MiniWoB++ [16] (top) and WorkArena [10] (bottom). MiniWoB consists of single-page simple tasks such as selecting a particular date and using a basic text editor, while WorkArena comprises multi-page complex tasks like filling forms and placing orders in an enterprise environment.

# F    Extended Related Work

**The Reproducibility Crisis in RL.**    The reproducibility crisis in large language models (LLMs) and reinforcement learning (RL) has garnered increasing attention, particularly due to the reliance on single seed results that distort the perceived performance of models. The reproducibility challenge* organized every year is a positive step towards addressing this. More concretely, Hochlehnert et al. [13] provide a critical examination of how such practices undermine the reliability of published findings, revealing that many reported gains are sensitive to implementation choices, such as random seeds and prompt formatting [13].

**Bandit-domain RLHF with LLMs.**    Previous work in RL for LLMs has predominantly focused on single-step tasks, which have shown effectiveness in mathematical reasoning and code generation [28, 9, 20]. While these approaches exhibit promising results, they are limited in their applicability to real-world scenarios, which often require multistep decision-making capabilities. The narrow focus

---

*https://reproml.org/

on bandit-style problems fails to address the complexities inherent in tasks that demand sequential interaction, highlighting a significant gap in the current research landscape.

**Interactive Agent Benchmarks.** To assess the capabilities of LLM agents in more realistic environments, benchmarks such as WebArena [29], WorkArena [10, 5], the Agent Company [27], and OSWorld [26] have been designed to evaluate agents on multi-step tasks across various domains. These benchmarks expose the limitations of current LLM agents, revealing that while they may perform well in controlled settings, their performance in practical applications remains subpar, underscoring the need for further advancements in agent robustness and generalization to multi-step planning.

**Best practices in deep RL.** Building on the recognition of reproducibility challenges and unstable RL training of LLM agents, recent studies have proposed best practices for training LLM agents using RL methods. Dang and Ngo [7] recommend leveraging high quality data, balancing easy and hard problems, and controlling length generation with cosine reward. Yu et al. [28] promote higher clipping in the GRPO loss to promote diversity and avoid entropy collapse, dynamic sampling to improve training efficiency and stability, token level gradients for long CoT sequences, and overlong reward shaping to reduce reward noise. Roux et al. [20] introduce tapered variant of importance sampling to speed up learning while maintaining stable learning dynamics. The proposed method (TOPR) allows the handling of both positive and negative examples in a fully offline setting. More generally, Hochlehnert et al. [13] emphasizes the need for greater methodological precision, particularly concerning decoding parameters, random seeds, prompt formatting, as well as the hardware and software frameworks, to guarantee transparent and thorough assessments of model performance.

**LLM Agents trained with RL on multi-step environments.** Recent advancements have sought to bridge the gap in training LLM agents for multi-step environments, with approaches like WebRL [19] and SWEET-RL [30] demonstrating significant progress. WebRL employs a self-evolving curriculum to address the challenges of sparse feedback and task scarcity, successfully enhancing the performance of open LLMs in web-based tasks [19]. Similarly, SWEET-RL introduces a hierarchical structure that enables effective credit assignment over multiple turns, improving policy learning and generalization in collaborative reasoning tasks [30]. These studies collectively illustrate the necessity of adapting RL techniques to accommodate the complexities of multi-step interactions, paving the way for more capable and versatile LLM agents.

# G   Test Set Hyper-Parameter Bootstrap Analysis

We overall find similar results between the held-out train and test tasks with respect to optimal hyper-parameters. While we see no large deviations, we find that some parameters such as curriculum learning from the instruct model and using error logs can have a larger beneficial effect on the held-out testing tasks.

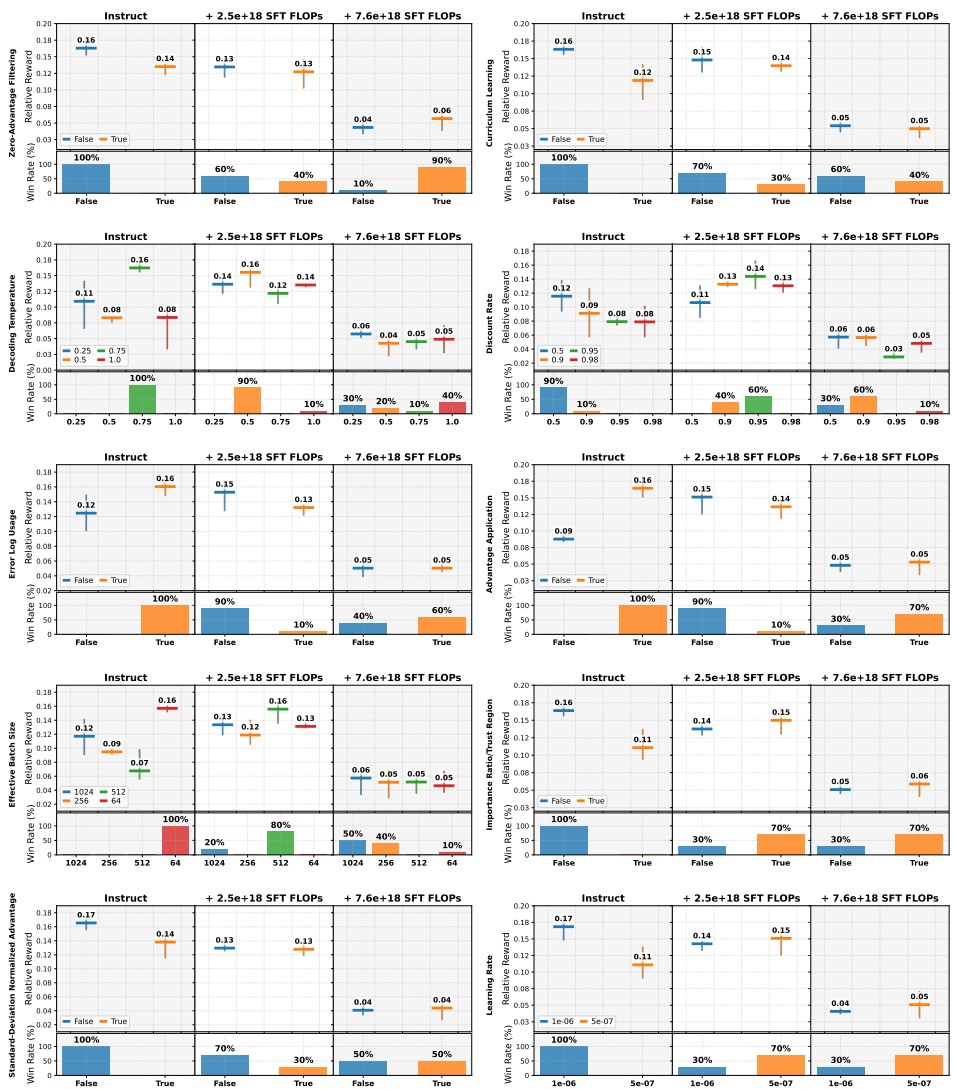

Figure 8: Bootstrap analysis ($n = 1000$ samples) of hyperparameter optimization across different SFT compute budgets on *test* held out tasks. Each subplot examines a different hyperparameter, including increasing SFT compute: the base instruct model (left), +2.5e+18 SFT FLOPs (middle), and +7.6e+18 SFT FLOPs (right). For each hyperparameter-compute combination, the top panel shows relative reward performance with error bars indicating 95% confidence intervals, while the bottom panel displays win rates representing the percentage of bootstrap iterations where each parameter value achieved maximum performance. Results demonstrate that optimal hyperparameter values often shift as model pre-training compute increases, suggesting that hyperparameter selection should be adapted based on the computational budget allocated to SFT.

## H Entropy

Figure 9 reports log-entropy as a function of training compute (FLOPs) across the training run.

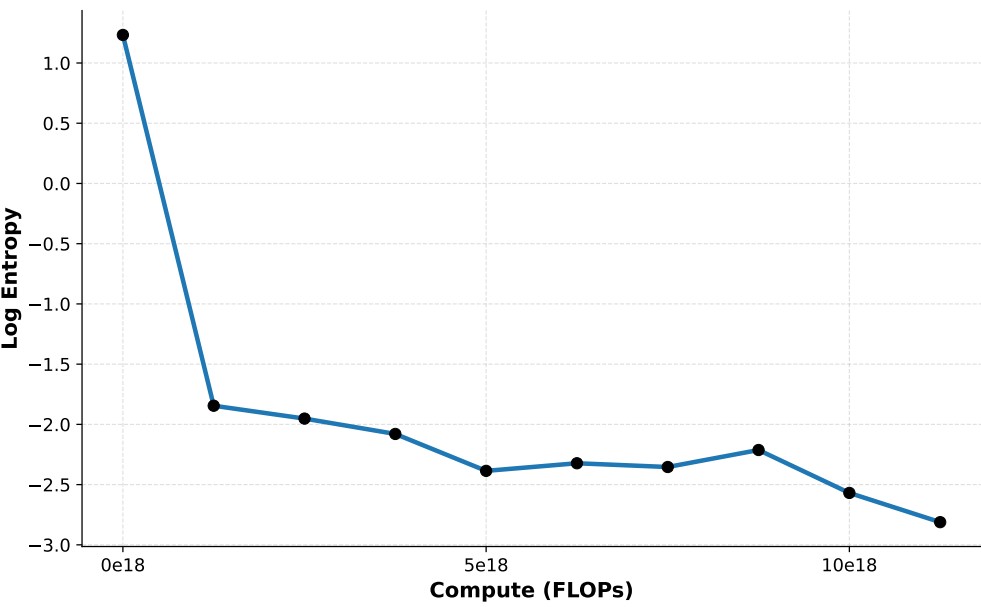

Figure 9: relationship between log-entropy and compute allocated to SFT warm-starting. We observe an initial sharp drop in log-entropy, followed by a period of stabilization and gradual decline. We speculate that this reduction in entropy, which makes rollouts more deterministic, may explain the difference in optimal hyperparameters between the base and warm-started checkpoints.

## I  Random Search Space

We conduct a random hyperparameter sweep over 1,370 training runs over the following parameter configurations:

- **Decoding temperature ($\rho_{\textbf{LLM}}$)**: Sampled from $\{0.1, 0.25, 0.5, 0.75, 1\}$
- **Curriculum learning**: Enabled or disabled (`True`, `False`)
- **Curriculum mean ($\mu_{\textbf{target}}$)**: $\{0.25, 0.5, 0.75\}$
- **Curriculum Temperature ($\rho_{\textbf{Curr}}$)**: $\{0.1, 0.3\}$
- **Discount rate**: $\{0.5, 0.8, 0.9, 0.95, 0.98, 1.0\}$
- **Grouped-relative advantage**: Enabled or disabled
- **Zero-advantage filtering**: Enabled or disabled
- **Standard-deviation normalized advantage**: Enabled or disabled
- **Effective batch size**: $\{64, 256, 512, 1024\}$
- **Learning rate**: $\{$1e-6, 5e-6, 5e-7$\}$
- **Error log feedback**: Enabled or disabled
- **Importance ratio**: Enabled or disabled

## J  Compute Allocation Hyperparameter Selection

For consistency, we select a single set of hyperparameters for all runs in Figure 1. To do so, we use the bootstrap algorithm described in Algorithm 1 where we use *all* the runs over all SFT checkpoints as input into `Results`. We then analyze the results of this bootstrap and use them to obtain the reported results.

The optimal parameters given by this aggregate bootstrap are:

- **Decoding temperature $\rho_{\textbf{LLM}}$**: 0.25

- **Curriculum learning**: `False`
- **Discount rate**: 0.90
- **Grouped-relative advantage**: `True`
- **Zero-advantage filtering**: `True`
- **Standard-deviation normalized advantage**: `True`
- **Effective batch size**: 512
- **Learning rate**: 1e-6
- **Error log feedback**: `True`
- **Importance Ratio**: `False`

## K  Correctness of the Bootsrap

One of the main motivations for using the boostrap algorithm for our hyper-parameter evaluation is the vast amount of literature showing its desrable properties for estimating the sampling distribution of data statistics.

The most important attribute for the robustness of the boostrap is its ability to exploit the Central Limit Theorem (CLT) to approximate the sampling distribution of a statistic. Using the CLT it can allow us to estimate its expectation and variability without parametric assumptions [12].

Specifically, in our use case, we want to evaluate, for each hyperparameter value $h$, how well it performs when paired with the best possible values of the other hyperparameters. The bootstrap algorithm yields an unbiased estimate for a target statistic given an initial $n$ samples (full runs in our case). Thus, we choose the statistic of interest to be the evaluation score $M$ conditioned on the hyper-parameter of interst $h$ assuming all other hyper-parameters $g$ are optimal:

$$T_h(D) = \max \left\{ M(h',g) : (h',g) \in D,\ h' = h \right\} \tag{16}$$

Bootstrapping this yields an empirical distribution over $T_h$ values, providing an unbiased estimate and confidence intervals for the maximum score achievable for $h$ when other hyperparameters are optimal.

Additionally, there exists results showing the rate of convergence and consistency of the bootstrap algorithm which we directly exploit. It is a well known result that as the number of training runs $n \to \infty$, the bootstrap estimate $\hat{T}_h$ of the maximum achievable score for hyperparameter value $h$ becomes consistent. That is:

$$\hat{T}_h \xrightarrow{p} T_h = \sup \left\{ M(h,g) : g \in \mathcal{G} \right\} \tag{17}$$

This follows from the law of large numbers [24], assuming sufficient coverage of other hyperparameters $\mathcal{G}$. The rate of convergence depends on the tail behavior of the conditional distribution $M \mid H = h$. Since our evaluation metric is bounded in $[0,1]$, the sub-Gaussian assumption holds automatically by Hoeffding's Lemma [14], ensuring concentration of the estimate. Under these conditions, the expected bias decays as $\mathcal{O}(1/n_h)$, where $n_h$ is the number of runs with $H = h$.

These are well-known results, which is why the bootstrap is an extremely useful and simple algorithm for estimating any statistic of interest [11, 12].

## L  Train and Test Splits Used in the Experiments

We evaluate generalization by training only on the *train* split and reporting performance on held-out tasks from the *test* split. For the held-out goals metric, we instantiate goal variations using seed ranges $[0, 2]$ [8] for training tasks (3 goals per task) and $[0, 9]$ [8] for test tasks (10 goals per task). Below we list the exact task identifiers used in our experiments for both benchmarks. Task names are the registry keys from the respective environments.

**WorkArena (Train: 24 tasks, Test: 7 tasks)**

**Train:**

- workarena.servicenow.all-menu
- workarena.servicenow.create-hardware-asset
- workarena.servicenow.create-incident
- workarena.servicenow.filter-asset-list
- workarena.servicenow.filter-change-request-list
- workarena.servicenow.filter-hardware-list
- workarena.servicenow.filter-service-catalog-item-list
- workarena.servicenow.filter-user-list
- workarena.servicenow.knowledge-base-search
- workarena.servicenow.order-apple-mac-book-pro15
- workarena.servicenow.order-development-laptop-p-c
- workarena.servicenow.order-ipad-mini
- workarena.servicenow.order-ipad-pro
- workarena.servicenow.order-sales-laptop
- workarena.servicenow.sort-change-request-list
- workarena.servicenow.sort-hardware-list
- workarena.servicenow.sort-incident-list
- workarena.servicenow.sort-service-catalog-item-list
- workarena.servicenow.sort-user-list
- workarena.servicenow.single-chart-value-retrieval
- workarena.servicenow.create-change-request
- workarena.servicenow.order-loaner-laptop
- workarena.servicenow.order-standard-laptop
- workarena.servicenow.create-problem

**Test:**

- workarena.servicenow.create-user
- workarena.servicenow.filter-incident-list
- workarena.servicenow.sort-asset-list
- workarena.servicenow.impersonation
- workarena.servicenow.order-apple-watch
- workarena.servicenow.order-developer-laptop
- workarena.servicenow.single-chart-min-max-retrieval

**MiniWoB++ (Train: 99 tasks, Test: 23 tasks)**

**Train:**

- miniwob.ascending-numbers
- miniwob.bisect-angle
- miniwob.book-flight
- miniwob.choose-date
- miniwob.choose-date-easy

- miniwob.choose-date-medium
- miniwob.choose-date-nodelay
- miniwob.choose-list
- miniwob.circle-center
- miniwob.click-button-sequence
- miniwob.click-checkboxes-soft
- miniwob.click-checkboxes-transfer
- miniwob.click-collapsible-2
- miniwob.click-collapsible-2-nodelay
- miniwob.click-collapsible-nodelay
- miniwob.click-color
- miniwob.click-dialog
- miniwob.click-dialog-2
- miniwob.click-link
- miniwob.click-menu
- miniwob.click-menu-2
- miniwob.click-scroll-list
- miniwob.click-shape
- miniwob.click-tab
- miniwob.click-tab-2
- miniwob.click-tab-2-hard
- miniwob.click-tab-2-medium
- miniwob.click-test
- miniwob.click-test-2
- miniwob.click-test-transfer
- miniwob.click-widget
- miniwob.copy-paste
- miniwob.copy-paste-2
- miniwob.count-shape
- miniwob.count-sides
- miniwob.daily-calendar
- miniwob.drag-box
- miniwob.drag-circle
- miniwob.drag-cube
- miniwob.drag-items
- miniwob.drag-items-grid
- miniwob.drag-shapes
- miniwob.drag-shapes-2
- miniwob.drag-sort-numbers
- miniwob.draw-circle
- miniwob.draw-line
- miniwob.email-inbox
- miniwob.email-inbox-delete
- miniwob.email-inbox-forward

- miniwob.email-inbox-forward-nl
- miniwob.email-inbox-forward-nl-turk
- miniwob.email-inbox-important
- miniwob.email-inbox-noscroll
- miniwob.email-inbox-reply
- miniwob.email-inbox-star-reply
- miniwob.enter-date
- miniwob.enter-text
- miniwob.enter-text-dynamic
- miniwob.enter-time
- miniwob.find-greatest
- miniwob.find-word
- miniwob.focus-text-2
- miniwob.form-sequence
- miniwob.form-sequence-2
- miniwob.generate-number
- miniwob.grid-coordinate
- miniwob.guess-number
- miniwob.highlight-text
- miniwob.hot-cold
- miniwob.identify-shape
- miniwob.login-user
- miniwob.login-user-popup
- miniwob.multi-layouts
- miniwob.multi-orderings
- miniwob.navigate-tree
- miniwob.odd-or-even
- miniwob.order-food
- miniwob.phone-book
- miniwob.read-table
- miniwob.read-table-2
- miniwob.resize-textarea
- miniwob.right-angle
- miniwob.scroll-text
- miniwob.scroll-text-2
- miniwob.search-engine
- miniwob.sign-agreement
- miniwob.simple-algebra
- miniwob.social-media
- miniwob.social-media-all
- miniwob.social-media-some
- miniwob.text-editor
- miniwob.text-transform
- miniwob.tic-tac-toe

- miniwob.use-autocomplete
- miniwob.use-autocomplete-nodelay
- miniwob.use-colorwheel
- miniwob.use-colorwheel-2
- miniwob.use-spinner
- miniwob.visual-addition

**Test:**

- miniwob.buy-ticket
- miniwob.click-button
- miniwob.click-option
- miniwob.click-pie-nodelay
- miniwob.drag-single-shape
- miniwob.email-inbox-nl-turk
- miniwob.enter-text-2
- miniwob.find-midpoint
- miniwob.focus-text
- miniwob.simple-arithmetic
- miniwob.stock-market
- miniwob.use-slider-2
- miniwob.click-checkboxes
- miniwob.click-checkboxes-large
- miniwob.click-collapsible
- miniwob.click-pie
- miniwob.click-shades
- miniwob.click-tab-2-easy
- miniwob.enter-password
- miniwob.form-sequence-3
- miniwob.highlight-text-2
- miniwob.unicode-test
- miniwob.use-slider

