# OpenReview forum: "How to Train Your LLM Web Agent: A Statistical Diagnosis"
_NeurIPS.cc/2025/Conference — NeurIPS 2025 poster_

### Official Review · Reviewer_Kh9s · 2025-07-01

**Clarity:** 1
**Significance:** 3
**Originality:** 4
**Rating:** 5
**Confidence:** 4

**Summary:**

The authors compare how to optimally allocate compute budgets on supervised fine-tuning (SFT) and reinforcement learning (RL, specifically GRPO) for fine-tuning LLM-based web agents on MiniWob++ and WorkArena.
They also do a large hyperparameter sweep and use bootstrapping to compare hyperparameter choices.
They find that it's worth spending compute on expensive teacher-generated traces for SFT before starting RL, but that adding RL will outperform pure SFT.

**Questions:**

1. Can you clarify how you use the Boltzmann distribution in curriculum learning?
2. Why is your test performance better than your training performance? For example, Llama-3.1-8b-SFT and Llama-3.1-8b-SFT+RL are all stronger on the WorkArena test set rather than the training set. Wouldn't we expect the training performance to be stronger, given that the model saw other seeds of the same tasks during training? This is true for Llama-3.1-8b-SFT and Llama-3.1-8b-RL on MiniWoB++ too.
3. Can you calculate how many trials were needed to get hyperparameters that were 90% as good as your final, optimal hyperparameters? That would be really helpful for understanding how many experiments other practioners should run in order to claim that they have near-optimal hyperparameters.
5. How do you measure AUC of success rates?
4. In Figure 4, right, why is the green line different to the blue line? Shouldn't SFT Warmup be the same as regular SFT? Also, what do the points/markers indicate in this graph?

**Ethical Concerns:**

["NO or VERY MINOR ethics concerns only"]

**Final Justification:**

The authors did additional experiments and clarified many of the confusing written aspects (graphs, tables, etc).

**Limitations:**

yes

**Paper Formatting Concerns:**

N/A. You misspelled practice as "prectice" in your related work (and in the appendix).

**Quality:**

2

**Strengths And Weaknesses:**

Strengths:

* The authors provide the first compute allocation curve. This graph (Figure 4, left) is awesome. It's a compelling and convincing visual argument that SFT+RL outperforms both pure SFT and pure RL. I cannot emphasize the value of this chart enough.
* The authors do extensive hyperparameter analysis, using bootstrapping to report confidence intervals, across various amounts of pre-RL SFT. I am really happy to see an extensive statistical analysis of web agents; as the authors correctly note, the field of RL+LLMs seems to be rediscovering many of the deep RL reproducibility challenges.

Weaknesses:

The biggest weakness is a lack of clarity in the writing.

* How is reward defined in web tasks? On line 76, it says $r_t \in [-1, 1]$. When does the model receive reward? Based on the writing around "sparse rewards", I assume it's only at the end of an episode. Does the model ever receive negative reward? Or is it really $r_t \in {0, 1}$?
* The curriculum learning via a Boltzmann distribution; what is the form of the Boltzmann distribution? I am not familiar with the distribution, and Wikipedia describes it as $p_i \propto \exp(-\frac{\epsilon_i}{kT})$ but the authors describe a target return $\mu_\text{target}$. How does the training algorithm select tasks?
* Lines 132-134: "Here the cost of $T_\text{RL}$ steps $F_\text{RL}(T_\text{RL})$ is equivalent to the sum of FLOPs for one data generation step and one training step of the student model." Do you mean the sum of flops for $T_\text{RL}$ data generation steps and $T_\text{RL}$ training steps for the student model?
* Line 237 says that 14 out of 33 WebArena tasks remain unsolved, but Figure 3 says 13/26 remain unsolved. Which is it?
* In Table 1, there are +/- numbers with no explanation. Are these standard deviations over the top 2 runs?

While I appreciate the statistical analysis of the hyperparameter sweep, I believe the conclusions are unlikely to generalize to other models or tasks due to the high variance in "conclusions". I would prefer the authors shift the messaging from "we used a statistical procedure and arrived at these general conclusions for LLM RL training" to "we used a statistical procedure and optimized our hyperparameters accordingly; here's how you can use the same procedure." Specifically, it would be really useful to see how many of those 240 runs were needed to arrive at the optimal hyperparameters.

---

> ### Author Rebuttal · Authors · 2025-07-31
>
> We thank the reviewer for their thoughtful and detailed review, as well as the positive comments regarding the value of our compute allocation curve and extensive statistical analysis. We address both the general improvements made since submission and each of the reviewer’s specific points below.
>
> ## General Comments
> Since the submission deadline, we have significantly expanded and strengthened our results:
>
> * **Expanded the hyperparameter search for the bootstrap experiment (from 240 to 1,370 runs):** We increased the size of the random hyperparameter sweep by more than 5x. Importantly, the main conclusions reported in the original submission remain unchanged, demonstrating the robustness of our findings. We believe these results showcase that the proposed bootstrap technique is robust and sample-efficient, effectively finding the same conclusions even when scaling the number of runs 5x. The only minor update is that using standard-deviation normalized advantage appears moderately better than not using it when starting RL from the instruct model.
>
> * **Generality across benchmarks:** We performed additional WorkArena experiments. Our updated results confirm that the main conclusion from MiniWoB++—that combining SFT and RL achieves the best absolute performance—also holds on WorkArena. We could also get some stable runs for Pure RL, improving the held-out task performance by 7.5%.
>
> * **Generality across model families:** We ran experiments on Qwen2.5 and replicated the main conclusion: SFT+RL is the only method able to match or exceed teacher performance. This provides evidence that our findings generalize beyond the LLaMA family of models.
>
> * **Bootstrap reliability:** We discuss the consistency and unbiasedness of the bootstrap estimator as well as the statistic we are specifically interested in estimating. Additionally, we provide examples of the reliability of our estimate in terms of standard error as the number of runs increases, showing that even under pessimistic assumptions we can maintain a relatively small standard error.
>
> ## Reviewer-Specific Comments
>
> *How is reward defined in web tasks? On line 76, it says . When does the model receive reward? Based on the writing around "sparse rewards", I assume it's only at the end of an episode. Does the model ever receive negative reward? Or is it really ?*
>
> The model receives a positive (+1) or negative (-1) upon the completion of the episode based on its usefulness. We are sorry about the confusion around “sparse” rewards. By sparse, we meant that the only reward that the model receives is at the end of the episode, i.e., no intermediate rewards for partially completing the tasks. We adjusted the paper to clarify this in the Experimental Setup’s part on benchmarks.
>
> *The curriculum learning via a Boltzmann distribution; what is the form of the Boltzmann distribution? I am not familiar with the distribution, and Wikipedia describes it as but the authors describe a target return . How does the training algorithm select tasks?*
>
> Boltzmann is just a softmax. The target return is the mode of the distribution, e.g., if target return is 50%, tasks close to 50% success rate will have the highest likelihood to be sampled by the algorithm. We clarified the manuscript accordingly.
>
> *Lines 132-134: "Here the cost of steps is equivalent to the sum of FLOPs for one data generation step and one training step of the student model." Do you mean the sum of flops for data generation steps and training steps for the student model?*
>
> The reviewer is right that the sentence was confusing and was right about the correction. We updated the manuscript.
>
> *Line 237 says that 14 out of 33 WebArena tasks remain unsolved, but Figure 3 says 13/26 remain unsolved. Which is it?*
>
> If you sum the 13/26 unsolved training tasks to the 1/6 unsolved test tasks, you get 14/33 unsolved tasks.
>
> *In Table 1, there are +/- numbers with no explanation. Are these standard deviations over the top 2 runs?*
>
> The reviewer is right, and we are sorry for this omission. The +/- refers to standard errors. In line 202 we indeed say that the average is the two top seeds.
>
> *While I appreciate the statistical analysis of the hyperparameter sweep, I believe the conclusions are unlikely to generalize to other models or tasks due to the high variance in "conclusions".*
>
> To address concerns about whether our conclusions generalize beyond the LLaMA family, we ran a very restricted hyperparameter random search of Qwen2.5 on MiniWoB++, taking into account the findings from the LLaMA bootstrap. We only swept over learning rate (LR), batch size (BS), and sampling temperature, fixing all other hyperparameters to those identified as optimal in the LLaMA experiments. From our sweep, we found that LR and BS are consistent with the results reported for LLaMA, with sampling temperature being higher at 0.75, which falls within the guidelines provided by the Qwen team.
> The updated MiniWoB++ results are shown below in the Rebuttal of Reviewer zHWC, DhYQ and jEdW, due to space constraints.
>
> We were able to replicate the MiniWoB++ result that SFT+RL achieves the best absolute performance with a new model family, namely Qwen2.5. It is also the only method able to surpass the teacher. Interestingly, we could not improve the performance of the instruct model using Pure RL in this case. After a few steps, the model often failed to follow the correct output format, began producing repeated tokens, or switched to unrelated languages. Nevertheless, the conclusions drawn from the LLaMA experiment still hold. We are actively working on the compute allocation study for Qwen2.5, which is compute-intensive. We will include these findings in the next version of the manuscript.
>
> *Can you clarify how you use the Boltzmann distribution in curriculum learning?*
>
> We thank the reviewers for bringing this point up, and we will clarify the formula in the paper to allow for better comprehension. To clarify here, the Boltzmann distribution is a well-known family of distributions that we are sure you are familiar with. In short, this distribution gives rise to the softmax and logistic loss functions traditionally used in classification. For simplicity, you may think that we sample from curriculum learning using a temperature-annealed softmax function:
> $$\frac{e^{\mu_{\text{task}}/\rho_{curr}}}{\sum_{\text{task}}e^{\mu_{\text{task}}/\rho_{curr}}}$$
> Thus, we calculate the probabilities for each task using this function and use them to sample tasks during training.
>
> *Why is your test performance better than your training performance? For example, Llama-3.1-8b-SFT and Llama-3.1-8b-SFT+RL are all stronger on the WorkArena test set rather than the training set. Wouldn't we expect the training performance to be stronger, given that the model saw other seeds of the same tasks during training? This is true for Llama-3.1-8b-SFT and Llama-3.1-8b-RL on MiniWoB++ too.*
>
> This is simply a feature of our random splits, which can randomly make the test set easier. For both benchmarks, the instruct model tends to do better on the test than on the train, but the tuned models close the gap, in line with the reviewer’s intuition.
>
> *Can you calculate how many trials were needed to get hyperparameters that were 90% as good as your final, optimal hyperparameters? That would be really helpful for understanding how many experiments other practitioners should run in order to claim that they have near-optimal hyperparameters.*
>
> Due to the CLT the standard error of the bootstrap estimate is bounded by a rate of $\mathcal{O}(1/\sqrt{n_h})$ thus we can make a comparison of optimality based on the standard error. For example, assuming a binary hyperparameter like applying advantage is evenly split across sampling groups, this gives us approximately $n_h = 120$ runs for each configuration from our initial 240 runs. To achieve a standard error that is only 10% less effective (i.e., 10% larger), we would need:
>  $$
> n' = \frac{n_h}{1.1^2} = \frac{120}{1.21} \approx 99
> $$
>
> Thus, approximately 99 runs are needed for this hyperparameter to obtain a confidence interval whose width is only 10% larger than with 120 runs.
>
> To further put this into perspective, since bootstrap exploits normality due to the CLT and since our scores are bounded in $[0,1]$, we can obtain performance bounds even under conservative variance assumptions.
>
> For instance, assume $\sigma = 0.25$ (which corresponds to a variance of 0.0625, and is still larger than typically observed in our experiments). Then:
>
> - With 100 runs:
>  $$
>  \text{SE}_{50} = \frac{0.25}{\sqrt{50}} = 0.035
>  $$
> - With 120 runs:
>  $$
>  \text{SE}_{120} = \frac{0.25}{\sqrt{120}} \approx 0.0228
>  $$
>
> Thus, even under the assumption of a high standard deviation in scores using even less than half our runs yields an estimate that is only ~15% worse than ours. While this may still be expensive for some labs, the purpose of our experiments are to take this into account and try to be able to provide robust recommendations.
>
> *How do you measure AUC of success rates?*
>
> This is clarified in the new version. Essentially, it is the mean success rate across all checkpoints of the run, serving as a proxy for overall training efficiency.
>
> *In Figure 4, right, why is the green line different to the blue line? Shouldn't SFT Warmup be the same as regular SFT? Also, what do the points/markers indicate in this graph?*
>
> This is because the SFT line is the best performing absolute run, which was not the one that learned the fastest (the one we branched off from for RL fine-tuning). We understand how this could be confusing. We are working on a better way to report some compute allocation insights on WorkArena for the next version.
>
> *N/A. You misspelled practice as "prectice" in your related work (and in the appendix).*
>
> Thanks! Fixed.

---

> > ### Comment · Reviewer_Kh9s · 2025-08-08
> >
> > Thank you for the clarifications and additional experiments. I have raised my score to a 5.

---

### Official Review · Reviewer_jEdW · 2025-07-03

**Clarity:** 2
**Significance:** 2
**Originality:** 2
**Rating:** 4
**Confidence:** 4

**Summary:**

This paper aims to fill the gap in reproducibility of training LLM agents with RL by providing a comprehensive training cookbook based on extensive experiments.
The cookbook includes suggestions on how to balance SFT and RL and also practical suggestions on hyper parameters selection.
To derive these insights, this paper fine-tunes Llama 3.1 8B and experiments with two agent benchmarks---MiniWoB++ and WorkArena. These two benchmarks are shipped with reward functions for RL training. To collect the trajectories for SFT, this paper directly uses a Llama 3.1 70B teacher model to generate them.
Concretely, this paper finds the following things: 1. balancing SFT and RL can be more compute-efficient than pure SFT and pure RL; 2. curriculum learning and error log feedback become counterproductive once SFT warmup is applied; 3. settings that can lead to better performance in GRPO, etc.

**Questions:**

1. I think it is important to be clear about how generalizable the conclusions derived in this paper are, given that the purpose of this paper is to provide a playbook for pushing open LLM agents' performance forward.
2. Could you justify the reliability of the bootstrapping method somehow? I am not very clear with your setting and may have some misunderstanding.
3. Why do you choose to use Llama 3.1 70B to generate the training data for SFT? In practice, high-quality human trajectories are probably still necessary. As a result, models trained with synthetic trajectories may not reveal the capacity of SFT models in production.

**Ethical Concerns:**

["NO or VERY MINOR ethics concerns only"]

**Final Justification:**

I have raised my overall evaluation after reading the response

**Limitations:**

Yes

**Quality:**

2

**Strengths And Weaknesses:**

### Strengths:
1. This paper has a clear motivation and tackles an important problem. The agent community wants to learn better practices of training LLM agents with RL.
2. Providing such a training cookbook is heavy and requires a ton of effort.


### Weaknesses:
1. Though this paper has already made considerable efforts in training models with 240 configurations, it may not be sufficient for the purpose of providing a comprehensive cookbook. Specifically, this paper only considers a very limited family of models (i.e., Llama 3.1 8B/70B as the student and teacher model respectively). The conclusions may not generalize to other models. For example, for models like Qwen 2.5, which has probably included many SFT data during continual pre-training, will we still need SFT before RL? Also, will the conclusions hold if we use a different model to generate the SFT data? What if we use human-crafted SFT data? In that case, should we still spend the same amount of compute on RL?
2. The bootstrapping method to obtain insights on hyper parameters may not be reliable. What if the original 240 configurations do not have a good coverage of good configurations?
3. The conclusions on different benchmarks are not consistent. As a result, this paper may fail to convey a generally meaningful message to the community. For example, on MiniWob++, SFT+RL outperforms RL-only, while on WorkArena it is a different story.

---

> ### Author Rebuttal · Authors · 2025-07-31
>
> We sincerely thank Reviewer jEdW for the thoughtful and constructive feedback, and for recognizing the importance of our work and the effort required to create a reproducible training cookbook for LLM agents. We address all concerns raised below.
>
> ## General comments
> Since the submission deadline, we have significantly expanded and strengthened our results:
>
> * **Expanded the hyperparameter search for the bootstrap experiment (from 240 to 1,370 runs):** We increased the size of the random hyperparameter sweep by more than 5x. Importantly, the main conclusions reported in the original submission remain unchanged, demonstrating the robustness of our findings. We believe these results showcase that the proposed bootstrap technique is robust and sample-efficient, effectively finding the same conclusions even when scaling the number of runs 5x. The only minor update is that using standard-deviation normalized advantage appears moderately better than not using it when starting RL from the instruct model.
>
> * **Generality across benchmarks:** We performed additional WorkArena experiments. Our updated results confirm that the main conclusion from MiniWoB++—that combining SFT and RL achieves the best absolute performance—also holds on WorkArena. We could also get some stable runs for Pure RL, improving the held-out task performance by 7.5%.
>
> * **Generality across model families:** We ran experiments on Qwen2.5 and replicated the main conclusion: SFT+RL is the only method able to match or exceed teacher performance. This provides evidence that our findings generalize beyond the LLaMA family of models.
>
> * **Bootstrap reliability:** We discuss the consistency and unbiasedness of the bootstrap estimator as well as the statistic we are specifically interested in estimating. Additionally, we provide examples of the reliability of our estimate in terms of standard error as the number of runs increases, showing that even under pessimistic assumptions we can maintain a relatively small standard error.
>
> ## Reviewer specific
>
> *1. Though this paper has already made considerable efforts in training models with 240 configurations, it may not be sufficient for the purpose of providing a comprehensive cookbook.*
>
> We have increased the number of runs from 240 to 1,370 and found that our main conclusions remain the same. Please see the general comments above for details.
>
> *2. The conclusions may not generalize to other models.*
>
> To address concerns about generality beyond the LLaMA family, we ran a very restricted hyperparameter random search of Qwen2.5 on MiniWoB++, taking into account the findings from the LLaMA bootstrap. We only swept over learning rate (LR), batch size (BS), and sampling temperature, fixing all other hyperparameters to those identified as optimal in the LLaMA experiments. From our sweep, we found that LR and BS are consistent with the results reported for LLaMA, with sampling temperature being higher at 0.75, which falls within the guidelines provided by the Qwen team.
> The updated MiniWoB++ results are shown below:
> | Model                           | Heldout Goals     | Heldout Tasks     |
> |---------------------------------|-------------------|-------------------|
> | Llama-3.3-70B Instruct (teacher)| 63.2 ± 2.4        | 61.9 ± 2.4        |
> | Llama-3.1-8B Instruct           | 29.5 ± 2.3        | 36.4 ± 2.4        |
> | Llama-3.1-8B SFT                | 53.4 ± 2.5        | 55.6 ± 2.5        |
> | Llama-3.1-8B RL                 | 43.5 ± 2.5        | 43.5 ± 2.5        |
> | Llama-3.1-8B SFT+RL             | **66.3 ± 2.4**    | **62.9 ± 2.4**    |
> |---------------------------------|-------------------|-------------------|
> | Qwen-2.5-72B Instruct (teacher) | 61.0 ± 3.4        | 59.0 ± 3.4        |
> | Qwen-2.5-7B Instruct            | 32.8 ± 3.3        | 37.0 ± 3.4        |
> | Qwen-2.5-7B SFT                 | 57.0 ± 3.5        | 56.5 ± 3.5        |
> | Qwen-2.5-7B RL                  | 32.8 ± 3.3        | 37.0 ± 3.4        |
> | Qwen-2.5-7B SFT+RL              | **63.5 ± 3.4**    | **62.0 ± 3.4**    |
>
> We replicated the MiniWoB++ result that SFT+RL achieves the best absolute performance with a new model family, namely Qwen2.5. It is also the only method able to surpass the teacher. Interestingly, Pure RL did not improve the instruct model in this case. After a few steps, the model often failed to follow the correct output format, began producing repeated tokens, or switched to unrelated languages. Nevertheless, the conclusions drawn from the LLaMA experiment still hold. We are actively working on the compute allocation study for Qwen2.5, which is compute-intensive, and will include these findings in the next version of the manuscript.
>
> *3. Also, will the conclusions hold if we use a different model to generate the SFT data?*
>
> We tried training LLaMA using Qwen as the teacher and vice versa. The SFT step still worked and gave comparable results. We have updated the paper with this finding.
>
> *4. What if we use human-crafted SFT data?*
>
> This is a great question, but it is outside the scope of this work, which focuses on how to optimally allocate compute/FLOPs between SFT and RL.
>
> *5. The bootstrapping method to obtain insights on hyperparameters may not be reliable. What if the original 240 configurations do not have good coverage of good configurations?*
>
> As discussed in the general comments, we increased the number of configurations to 1,370 and found that the main conclusions remained unchanged. This increase in coverage further demonstrates the robustness of our bootstrap-based analysis.
>
> We will further clarify the reliability of our technique. The bootstrap leverages the Central Limit Theorem (CLT) to approximate the sampling distribution of a statistic, allowing us to estimate its expectation and variability without parametric assumptions [Efron & Tibshirani, 1993].
>
> **Goal:** We want to evaluate, for each hyperparameter value $h$, how well it performs when paired with the best possible values of the other hyperparameters. The bootstrap algorithm yields an unbiased estimate for a target statistic given an initial $n$ samples (full runs in our case). Thus, we choose the statistic of interest to be the evaluation score $M$ conditioned on the hyper-parameter of interest $h$ assuming all other hyper-parameters $g$ are optimal:
>
> $$
> T_h(D) = \max \{ M(h', g) : (h', g) \in D, h' == h \}$$
>
> Bootstrapping this yields an empirical distribution over $T_h$ values, providing an unbiased estimate and confidence intervals for the maximum score achievable for $h$ when other hyperparameters are optimal.
>
> **Optimality and Reliance:**
>
> As the number of training runs $n \to \infty$, the bootstrap estimate $\hat{T}_h$ of the maximum achievable score for hyperparameter value $h$ becomes consistent. That is:
>
> $$
> \hat{T}_h \xrightarrow{p} T_h = \sup \{ M(h, g) : g \in \mathcal{G} \}
> $$
>
> This follows from the law of large numbers [Wainwright, 2019], assuming sufficient coverage of other hyperparameters $G$. The rate of convergence depends on the tail behavior of the conditional distribution $M \mid H = h$. Since our evaluation metric is bounded in $[0, 1]$, the sub-Gaussian assumption holds automatically by Hoeffding’s Lemma [Hoeffding, 1963; Wainwright, 2019], ensuring concentration of the estimate. Under these conditions, the expected bias decays as $\mathcal{O}(1 / n_h)$, where $n_h$ is the number of runs with $H = h$. All of these are well known results thus why the bootstrap algorithm is an extremely useful and simple algorithm to use to estimate any statistic of interest [Efron, 1979; Efron & Tibshirani, 1993].
>
>
> **References:**
> - Efron, B. (1979). Bootstrap methods: Another look at the jackknife. *The Annals of Statistics*, 7(1), 1-26.
> - Efron, B., & Tibshirani, R. J. (1993). *An Introduction to the Bootstrap*. Chapman & Hall.
> - Hoeffding, W. (1963). Probability inequalities for sums of bounded random variables. *Journal of the American Statistical Association*, 58(301), 13-30.
> - Wainwright, M. J. (2019). *High-Dimensional Statistics: A Non-Asymptotic Viewpoint*. Cambridge University Press. (See Chapter 2 on concentration inequalities.)
>
>
> *6. The conclusions on different benchmarks are not consistent.*
>
> In the deadline version of the manuscript, SFT+RL outperformed RL-only in both benchmarks. In our updated results, SFT+RL remains the best approach in terms of absolute performance for both benchmarks and both model families.
>
> *7. Why do you choose to use LLaMA 3.1 70B to generate the training data for SFT?*
>
> It does not scale as easily to use human-generated data; a compute allocation study would be infeasible in that setting. In practice, one often has access to a larger model that can collect higher-quality data to train a smaller, more efficient student model.
> Scaling the offline training using human-generated traces is difficult, expensive, and time-consuming. We specifically restricted ourselves to the LLaMA family of models as they are open-source and there is no legal liability in using their traces to train other models. Benchmarks like WorkArena require domain knowledge and high-quality reasoning rationales, and prior work [1,2,3] shows that model-generated rationales can match or even exceed human rationales in some tasks. Since our student model is LLaMA 3.1 8B, we chose LLaMA 3.1 70B as the teacher.
>
> 1. Gulcehre, Caglar, et al. "Reinforced self-training (rest) for language modeling." arXiv preprint arXiv:2308.08998 (2023).
> 2. Agarwal, Rishabh, et al. "Many-shot in-context learning." Advances in Neural Information Processing Systems 37 (2024): 76930-76966.
> 3. Wang, Yizhong, et al. "Self-instruct: Aligning language models with self-generated instructions." arXiv preprint arXiv:2212.10560 (2022).

---

### Official Review · Reviewer_zHWC · 2025-07-12

**Clarity:** 3
**Significance:** 2
**Originality:** 3
**Rating:** 4
**Confidence:** 4

**Summary:**

This paper presents a statistical analysis of hyperparameter selection for fine-tuning the LLaMA3-8B model into a web agent. The training pipeline includes supervised fine-tuning (SFT) via distillation from LLaMA3-70B, followed by GRPO optimization. The paper suggest to (1) Branching into RL early (2) an optimal hyper-parameter configuration worked for training LLaMA3-8B into a web-agent (with temperature=0.25, no curriculum learning, zero-advantage filtering, etc.), (3) the paper propose a strategy for bootstrapping estimation of the important hyperparameter.

**Questions:**

See weakness

**Ethical Concerns:**

["NO or VERY MINOR ethics concerns only"]

**Final Justification:**

I will raise the score to boardline accept, as most of my concerns have been addressed in the authors' rebuttal

**Limitations:**

See weakness

**Paper Formatting Concerns:**

No format issue

**Quality:**

2

**Strengths And Weaknesses:**

# Strength:
1. The paper is clearly written and easy to follow.
2. The proposed hyperparameter selection strategy is both efficient and potentially scalable for use in LLM agent training.
3. The experiments, including ablation studies, are thorough and well-executed.


# Weakness:
1. Figure 2 is unclear. The task described in the figure is confusing and would benefit from a more detailed demonstration.
2. All hyperparameter-choosing experiments are conducted exclusively on the LLaMA model and also all the corresponding insights (e.g. SFT + RL > RL alone, etc). It raises concerns about the generalizability of the findings to other LLMs, such as Qwen, DeepSeek, etc or other model size (1.5B, 32B, etc)
3. The insight that “SFT + RL > RL alone” seems straightforward, but I am curious about the underlying reasons. Specifically:
 (1) Have you tried using test-time scaling inference to mitigate the Pass@1 bias in evaluation?
 (2) Why does pure RL perform worse than its SFT and SFT+RL counterparts? For example, in WorkArena (Table 1), RL applied to LLaMA3.1-8B-RL achieves a score of 0.0—worse than the base model. Is this because the web-agent task demands a specific reasoning format, such that without the prior structure introduced by SFT, rollouts receive zero reward and RL fails to learn anything meaningful? If so, it would be helpful for the authors to provide a qualitative analysis explaining this behavior.
 4. I find the role of error log feedback unclear. Is error feedback incorporated during SFT/RL training in prompt, or is it only used at test time?
 5. Why is a decoding temperature of 0.25 optimal for this task? This seems counterintuitive, as such a low temperature could lead to deterministic decoding and reduce rollout diversity. Could this explain the zero improvement of pure RL in WorkArena?
 6. The hyperparameter configuration is also somewhat unclear. While Appendices E–F describe the search space, it's not obvious whether key hyperparameters—such as batch size and learning rate—are shared between the RL and SFT phases.

---

> ### Author Rebuttal · Authors · 2025-07-31
>
> We sincerely thank Reviewer zHWC for the thoughtful and constructive feedback, and for recognizing the clarity of writing, the scalability of our hyperparameter selection strategy, and the thoroughness of our experiments and ablations. We are encouraged by these positive remarks and address all concerns raised below.
>
> ## General comments
> Since the submission deadline, we have significantly expanded and strengthened our results:
>
> * **Expanded the hyperparameter search for the bootstrap experiment (from 240 to 1,370 runs):** We increased the size of the random hyperparameter sweep by more than 5x. Importantly, the main conclusions reported in the original submission remain unchanged, demonstrating the robustness of our findings. We believe these results showcase that the proposed bootstrap technique is robust and sample-efficient, effectively finding the same conclusions even when scaling the number of runs 5x. The only minor update is that using standard-deviation normalized advantage appears moderately better than not using it when starting RL from the instruct model.
>
> * **Generality across benchmarks:** We performed additional WorkArena experiments. Our updated results confirm that the main conclusion from MiniWoB++—that combining SFT and RL achieves the best absolute performance—also holds on WorkArena. We could also get some stable runs for Pure RL, improving the held-out task performance by 7.5%.
>
> * **Generality across model families:** We ran experiments on Qwen2.5 and replicated the main conclusion: SFT+RL is the only method able to match or exceed teacher performance. This provides evidence that our findings generalize beyond the LLaMA family of models.
>
> * **Bootstrap reliability:** We discuss the consistency and unbiasedness of the bootstrap estimator as well as the statistic we are specifically interested in estimating. Additionally, we provide examples of the reliability of our estimate in terms of standard error as the number of runs increases, showing that even under pessimistic assumptions we can maintain a relatively small standard error.
>
>
> ## Reviewer specific
> *1. Figure 2 is unclear. The task described in the figure is confusing and would benefit from a more detailed demonstration.
> We understand that it might be difficult to get the gist of both benchmarks in that single figure.*
>
> We extended the Appendix with examples of tasks for both benchmarks and intend to extend the main part of the paper upon acceptance using the extra allowed page.
>
> *2. All hyperparameter-choosing experiments are conducted exclusively on the LLaMA model and also all the corresponding insights (e.g. SFT + RL > RL alone, etc). It raises concerns about the generalizability of the findings to other LLMs (...).*
>
> To address concerns about generality beyond the LLaMA family, we ran a very restricted hyperparameter random search of Qwen2.5 on MiniWoB++, taking into account the findings from the LLaMA bootstrap. We only swept over learning rate (LR), batch size (BS), and sampling temperature, fixing all other hyperparameters to those identified as optimal in the LLaMA experiments. From our sweep, we found that LR and BS are consistent with the results reported for LLaMA, with sampling temperature being higher at 0.75, which falls within the guidelines provided by the Qwen team.
> The updated MiniWoB++ results are shown below:
>
> | Model                           | Heldout Goals     | Heldout Tasks     |
> |---------------------------------|-------------------|-------------------|
> | Llama-3.3-70B Instruct (teacher)| 63.2 ± 2.4        | 61.9 ± 2.4        |
> | Llama-3.1-8B Instruct           | 29.5 ± 2.3        | 36.4 ± 2.4        |
> | Llama-3.1-8B SFT                | 53.4 ± 2.5        | 55.6 ± 2.5        |
> | Llama-3.1-8B RL                 | 43.5 ± 2.5        | 43.5 ± 2.5        |
> | Llama-3.1-8B SFT+RL             | **66.3 ± 2.4**    | **62.9 ± 2.4**    |
> |---------------------------------|-------------------|-------------------|
> | Qwen-2.5-72B Instruct (teacher) | 61.0 ± 3.4        | 59.0 ± 3.4        |
> | Qwen-2.5-7B Instruct            | 32.8 ± 3.3        | 37.0 ± 3.4        |
> | Qwen-2.5-7B SFT                 | 57.0 ± 3.5        | 56.5 ± 3.5        |
> | Qwen-2.5-7B RL                  | 32.8 ± 3.3        | 37.0 ± 3.4        |
> | Qwen-2.5-7B SFT+RL              | **63.5 ± 3.4**    | **62.0 ± 3.4**    |
>
>
> We replicated the MiniWoB++ finding that SFT+RL achieves the best absolute performance with a new model family, namely Qwen2.5. It is also the only method able to surpass the teacher. Interestingly, Pure RL did not improve the instruct model in this case: after a few steps, the model often failed to follow the correct output format, produced repeated tokens, or switched to unrelated languages. Nevertheless, the conclusions drawn from the LLaMA experiment still hold. We are actively working on the compute allocation study for Qwen2.5 and will include these findings in the next version of the manuscript.
>
> *3. “SFT + RL > RL alone” seems straightforward. (1) Have you tried using test-time scaling inference to mitigate the Pass@1 bias in evaluation? (2) Why does pure RL perform worse than its SFT and SFT+RL counterparts? For example, in WorkArena (Table 1), RL applied to LLaMA3.1-8B-RL achieves a score of 0.0—worse than the base model.*
>
> It is not straightforward that SFT+RL forms a better compute Pareto front, given that expert data collection is much more costly (9x in our case). Regarding Pass@1 bias, there should be no bias because we model-select using Pass@1 and report Pass@1 on the test set. We are happy to run additional tests for the camera-ready version if the reviewer insists.
> On Pure RL performance: the main issue is that RL applied to the instruct model struggles to operate effectively on challenging WorkArena tasks (starting at ~8% on training tasks). This results in numerous errors, including parsing issues, verbose rationales, clicking non-existent elements, and failure to follow output templates. We allocated additional compute to Pure RL on WorkArena and observed a 7.5% improvement on the held-out test tasks.
> We also scaled up WorkArena experiments post-deadline and fixed environment issues where models could break WorkArena instances using admin rights to change the UI, leading to infeasible tasks. This allowed us to scale the experiments more effectively. Our updated results confirm that SFT+RL remains the best approach on WorkArena. Note that we changed the task split after upgrading to a newer version of WorkArena, which explains some discrepancies with earlier results.
> The updated WorkArena results are shown below:
>
> | Model                           | Heldout Goals   (%)  | Heldout Tasks  (%)   |
> |--------------------------------|--------------------|-------------------|
> | Llama-3.3-70B Instruct (teacher)	| 36.0 ± 2.4         	| 44.0 ± 2.5 |
> | Llama-3.1-8B Instruct           	| 8.0 ± 1.0          	| 4.0 ± 1.0 |
> | Llama-3.1-8B SFT                	| 28.4 ± 2.3         	| 26.4 ± 2.2 |
> | Llama-3.1-8B RL                 	| 8.0 ± 1.9          	| 11.5 ± 2.3 |
> | **Llama-3.1-8B SFT+RL**         	| **34.6 ± 2.4**     	| **28.0 ± 2.2** |
>
> These results show that the SFT+RL approach continues to dominate in absolute performance, while Pure RL can achieve modest improvements with additional tuning.
>
> *4. I find the role of error log feedback unclear. Is error feedback incorporated during SFT/RL training in prompt, or is it only used at test time?*
>
> Error log feedback means that if the model encounters an environment error (e.g., clicking on a non-existent item), we return the error log to the model in the next step within its prompt. This is used during training.
>
> *5. Why is a decoding temperature of 0.25 optimal for this task? Could this explain the zero improvement of pure RL in WorkArena?*
>
> The sampling temperature largely depends on the initial entropy of the model. While 0.25 may be low for some models, it falls within the recommended range for the LLaMA 3 family. The lack of Pure RL improvement in WorkArena was primarily due to environment errors, which we have since alleviated, as reflected in the updated results above.
>
> *6. The hyperparameter configuration is also somewhat unclear. Are key hyperparameters shared between the RL and SFT phases?*
>
> We tune the hyperparameters for both SFT and RL phases. RL has considerably more hyperparameters to tune, so we focus there. SFT is more stable and works well with a wider range of settings. Specifically, we found that a learning rate of 1e-6 and batch size of 512 are generally good defaults for SFT and are also used in RL.

---

> > ### Comment · Reviewer_zHWC · 2025-08-07
> >
> > Thanks for your response, most of my concerns have been addressed. I would like to see more clear explanation and more thorough experiments on the paper in the next version. (e.g., the Qwen experiment).
> >
> > Thus I will raise my score.

---

### Official Review · Reviewer_8JSX · 2025-07-13

**Clarity:** 2
**Significance:** 2
**Originality:** 2
**Rating:** 5
**Confidence:** 4

**Summary:**

The paper provides guidelines and a recipe for training an LLM web agent with RL for compute-constrained practitioners to use, focusing on two benchmarks: MiniWoB++ and WorkArena. In the recipe, a larger teacher model generates reasoning traces for the agentic web task, which are then filtered for correctness and used to train a smaller student model with SFT. The student model then uses the rest of the compute budget to RL train itself on the task using GRPO. The authors provide guidelines for how much SFT to perform versus GRPO, enabling practitioners to strike a Pareto optimal balance. The authors also perform a statistical hyperparameter sensitivity study to recommend appropriate hyperparameters to use for this process.

**Questions:**

- Why is there an asymmetry between the two charts of Figure 4? I would like to see the analysis of WorkArena be at the level of the analysis on MiniWoB++. In particular, I would like to see the branching timing into RL training studied.
- Why is RL training failing to be effective on WorkArena; in particular, why does standard RL training get 0% accuracy? I would like to see an adequate explanation and/or improvement in this area.
- What is the effectiveness of the Boltzmann sampling strategy? How does it compare to a uniform baseline? If it is more effective, I would like to see a breakdown of what makes it more effective.
- Is this recipe generalizable to models outside the Llama family? What about the hyperparameters? Evidence of this would support the generalizability of the method.

**Ethical Concerns:**

["NO or VERY MINOR ethics concerns only"]

**Final Justification:**

I thank the authors for clearly addressing my concerns, and as such I am increasing my score. The most important factors for this were the addition of working experiments on WorkArena, which I find is critical, and demonstrating that the recipe generalizes to other model families. Together, these mostly address my concerns of generality. I do believe that at least a small version of the branching study (e.g. localized around where MiniWoB++ was optimal, since compute is expensive) would be beneficial for the final version of this paper, to see how consistent that is across tasks. This will help to maximize the impact of the paper.

**Limitations:**

yes

**Quality:**

2

**Strengths And Weaknesses:**

# Strengths
## Quality
- The experiments and ablations for MiniWoB++ go into an effective level of depth about choices to make during training web agents. In particular, the experiment design for when to branch out of SFT training is well-constructed.
- The author's method of performing a statistical hyperparameter analysis appears effective for this resource constrained setting on aggregating information about the senisitivity of different hyperparameter choices.
## Clarity
- Many of the figures effectively communicate the results, such as the hyperparameter sensitivity analysis, and the left chart of Figure 4.
## Significance
- Considering the popularity of training web agents, providing a guideline for resource-constrained teams to train a successful agent is highly valuable to the community. In particular, the information on when to branch out to perform RL training will be valuable for teams to save their resources.
## Originality
- The authors' method of performing the statistical hyperparameter sensitivity analysis appears to be novel.
- The Boltzmann sampling strategy for training curriculum appears to be novel.

# Weaknesses
## Quality
- The RL training fails to improve model performance on WorkArena, with RL alone dropping train/test held-out performance from 9%/5% accuracy to 0% accuracy each, respectively. The drop to zero accuracy without an accompanying analysis indicates that either something may have been wrong with the training environment, or the training recipe does not generalize to benchmarks other than MiniWoB++. Either way, the generality of this training recipe has not been demonstrated, having only been successful on one benchmark.
- The effects of the Boltzmann sampling strategy for training curriculum are not compared to any baseline, with no evidence provided about its effectiveness compared to uniform sampling.
## Clarity
- In Section 6: Ablation and Sensitivity Analysis, the hyperparameter analysis does not go into depth, and often states results without well-substantiated explanations. For example on lines 257-259, the paper states "relative advantage helps only after SFT, while using raw rewards works better when starting directly from the Instruct model, possibly due to how advantage scaling interacts with the initial weights", without explaining the basis behind the claim in the later half of the sentence.
- The left and right charts of Figure 4 appear to represent the same analysis on the two different benchmarks, yet the left chart (for MiniWoB++) appears much more fleshed out than the right chart (for WorkArena). Additionally, there appears to be an asymmetry in the experiments with no explanation: why is there only one branch out of the SFT training curve for WorkArena, and why is the curve for the SFT warm-up different from the standard SFT curve, are different hyperparameters used?
## Significance
- The generality of these guidelines may be limited; the authors did not study whether their results generalize to new models or different compute budgets. Additionally, It is unclear whether the positive results are generalizable beyond MiniWoB++.
## Originality
- While appearing to be novel, the effectiveness of the Boltzmann sampling strategy on the training curriculum is not studied.

---

> ### Author Rebuttal · Authors · 2025-07-31
>
> We sincerely thank Reviewer 8JSX for the thoughtful and constructive feedback, and for recognizing the significance of our . We are encouraged by the positive assessment regarding our reproducibility focus and overall impact on the LLM web agent training subfield and have worked extensively to address your concerns.
>
> ## General comments
>
> Since the submission deadline, we have significantly expanded and strengthened our results:
> * **Expanded the hyperparameter search for the bootstrap experiment (from 240 to 1,370 runs):** We increased the size of the random hyperparameter sweep by more than 5x. Importantly, the main conclusions reported in the original submission remain unchanged, demonstrating the robustness of our findings. We believe these results showcase that the proposed bootstrap technique is robust and sample-efficient, effectively finding the same conclusions even when scaling the number of runs 5x. The only minor update is that using standard-deviation normalized advantage appears moderately better than not using it when starting RL from the instruct model.
>
> * **Generality across model families:** We ran experiments on Qwen2.5 and replicated the main conclusion: SFT+RL is the only method able to match or exceed teacher performance. This provides evidence that our findings generalize beyond the LLaMA family of models.
>
> * **Generality across benchmarks:** We performed additional WorkArena experiments. Our updated results confirm that the main conclusion from MiniWoB++—that combining SFT and RL achieves the best absolute performance—also holds on WorkArena. We could also get some stable runs for Pure RL, improving the held-out task performance by 7.5%.
>
> These results show that the SFT+RL approach continues to dominate in absolute performance, while Pure RL can achieve modest improvements with additional tuning. We will include this updated table and analysis in the camera-ready version.
>
> ## Reviewer-specific comments
>
> *1. "The RL training fails to improve model performance on WorkArena, with RL alone dropping train/test held-out performance from 9%/5% accuracy to 0% accuracy each, respectively. "*
>
> We performed additional experiments on WorkArena to address the reviewers’ concerns about generalizability. We were able to run many more WorkArena experiments after the deadline. We also fixed environment issues related to models breaking the WorkArena instances by using their admin rights to change the UI, which could lead to infeasible tasks. This allowed us to scale the experiment much more effectively.
> Our updated results confirm that the main conclusion from MiniWoB++—that combining SFT and RL achieves the best absolute performance—also holds on WorkArena. We could also obtain stable runs for Pure RL, improving the held-out task performance by 7.5%. Note that since the deadline, we changed the task split because we upgraded to a newer version of WorkArena, which explains some of the discrepancies with earlier results.
> We have addressed this issue—our initial runs failed due to environment-related bugs that caused instability and crashes. Since fixing these issues, we have obtained stable RL runs and now report the updated results below.
>
>
> Due to character limits, we cannot insert the full table here, please see our response to reviewer `zHWC` for the full results.
>
>
> *"2. Either way, the generality of this training recipe has not been demonstrated, having only been successful on one benchmark."*
>
> To address generality, we ran experiments with the Qwen2.5 model family. We performed a very restricted hyperparameter random search on MiniWoB++, only sweeping over learning rate, batch size, and sampling temperature (all other hyperparameters were fixed to those identified as optimal in the LLaMA bootstrap). The updated MiniWoB++ results are shown below:
>
> As above, due to character limits, we cannot insert the full table here, please see our response to reviewer `zHWC` for the full results.
>
> We were able to replicate the MiniWoB++ result that SFT+RL achieves the best absolute performance with a new model family, namely Qwen2.5. It is also the only method able to surpass the teacher. Interestingly, we could not improve the performance of the instruct model using Pure RL in this case. After a few steps, the model often failed to follow the correct output format, began producing repeated tokens, or switched to unrelated languages. Nevertheless, the conclusions drawn from the LLaMA experiment still hold.
>
> *3. The effects of the Boltzmann sampling strategy for training curriculum are not compared to any baseline, with no evidence provided about its effectiveness compared to uniform sampling.*
>
> In our ablation study, we compare curriculum learning via Boltzmann sampling against uniform sampling (which is what we use when curriculum learning is disabled). We find Boltzmann sampling helps when training from instruct models, where frequent failures can degrade performance. After SFT, however, curriculum learning tends to hurt, likely because the model already performs well on most tasks. In this setting, curriculum sampling causes the model to over-focus on the few very hard tasks, while continuing to train on tasks it already performs well on is still beneficial.
> While we do not claim to introduce this sampling strategy, our goal is to empirically investigate which hyperparameters and training techniques matter and how their effects vary across compute budgets. We agree that optimal curriculum structure remains an open question, being an active research area (e.g., in active inference)
>
>
> *"4. The generality of these guidelines may be limited; the authors did not study whether their results generalize to new models or different compute budgets."*
>
> We agree that further generalization is important, which is why we have now included results using Qwen2.5 (see above), where SFT+RL again achieves the best absolute performance. We are also actively conducting compute allocation experiments for Qwen2.5.
> Regarding compute budgets, we are unsure what the reviewer is referring to. Our compute allocation study already explores how to distribute a fixed compute budget across SFT, RL, and SFT+RL. If the reviewer meant something else, we would appreciate further clarification.
>
>
> *"5. In Section 6: Ablation and Sensitivity Analysis, the hyperparameter analysis does not go into depth, and often states results without well-substantiated explanations. For example on lines 257-259, the paper states "relative advantage helps only after SFT, while using raw rewards works better when starting directly from the Instruct model, possibly due to how advantage scaling interacts with the initial weights", without explaining the basis behind the claim in the later half of the sentence."*
>
> We appreciate the reviewer pointing this out and will revise the appendix to include deeper discussion. Our initial reasoning was that standard advantage normalization could scale updates pathologically, especially early in training, where naive reinforce (bounded in [-1, 1]) may be more stable.
> Upon further analysis, we believe the main factor is the interaction between reward baselines and the model's initial capabilities. Specifically, when starting from an instruct model, the baseline often cancels gradient updates on tasks the model performs well. Yet we find that positive updates even on such tasks are important for stabilizing learning. In contrast, SFT-trained models already acquire key skills (e.g., formatting, parsing HTML/AXTrees), so continuing to train on tasks they already solve yields diminishing returns. We believe this more accurately explains the advantage of naive reinforce in that setting.
>
>
> *"6. The left and right charts of Figure 4 appear to represent the same analysis on the two different benchmarks, yet the left chart (for MiniWoB++) appears much more fleshed out than the right chart (for WorkArena). Additionally, there appears to be an asymmetry in the experiments with no explanation: why is there only one branch out of the SFT training curve for WorkArena, and why is the curve for the SFT warm-up different from the standard SFT curve, are different hyperparameters used?"*
>
> We agree this presentation could be confusing. We did not repeat the full compute allocation study for WorkArena due to resource constraints—it was meant to test whether MiniWoB++ conclusions hold in a different setting. We have clarified the manuscript to explain that WorkArena results are intended to validate the generality of the MiniWoB++ findings. We also revised the figure to use a format consistent with the MiniWoB++ version.
>
> *"7. The generality of these guidelines may be limited...."*
>
> As noted above, we extended our experiments to WorkArena and to the Qwen2.5 model family, demonstrating that our main conclusions (e.g., that SFT+RL provides the best results) generalize across models and benchmarks.
>
> *Why is there an asymmetry between the two charts of Figure 4? I would like to see the analysis of WorkArena be at the level of the analysis on MiniWoB++. In particular, I would like to see the branching timing into RL training studied.*
>
> We agree this would be valuable, but running a full compute allocation study on WorkArena would be very costly given the instability and runtime of its RL training. If this analysis is essential, we will commit to including it upon acceptance.
>
> *"8. Is this recipe generalizable to models outside the Llama family? What about the hyperparameters?."*
>
> Please see General Comment Section 2, where we describe how the hyperparameters identified via the bootstrap for LLaMA transfer successfully to Qwen2.5. This supports the generality of both the training recipe and hyperparameter recommendations.
>
>
> We thank the reviewer for all their feedback which has greatly improved our work as a result. If there are any further clarifications we are happy to engage in further discussion.

---

> ### Comment · Reviewer_8JSX · 2025-08-07
>
> Thank you for addressing my concerns, I believe this greatly strengthens the quality and impact of the paper, and my score has been updated. About point 7, I believe at least a small version of this experiment would be very beneficial for the final paper.

---

> > ### Author Response · Authors · 2025-08-08
> > **Response to Reviewer 8JSX – Branching Timing Experiment on WorkArena**
> >
> > We sincerely thank Reviewer 8JSX for the follow-up and for updating their score. Given the reviewer’s continued emphasis on the value of including at least a small-scale version of the branching timing experiment on WorkArena (Point 7), we will run this experiment and include the results in the camera-ready version.

---

### Official Review · Reviewer_DhYQ · 2025-07-15

**Clarity:** 4
**Significance:** 3
**Originality:** 1
**Rating:** 5
**Confidence:** 3

**Summary:**

LLM agent training is a field that has been substantially hindered by reproducibility issues and lack of transparency in both the RLHF-stage and the Agent-stage. This paper provides a statistically driven study of training LLM agents for web tasks.

**Questions:**

1. Some other works I've read seem to present mutually contradictory accounts of how important SFT before RLHF is. The general vibe I've gotten is that it may depend on the base model. Do you have any thoughts on this? Do you have intuition for if you'll see similar results if you use DeepSeek or Qwen?
2. A key issue for deployment is the robustness of a model across a wide diversity of website structures. Arguably a 1% failure rate for a general purpose web agent is problematically high. Have you considered this or do you plan to assess this?

**Ethical Concerns:**

["NO or VERY MINOR ethics concerns only"]

**Final Justification:**

I am already strongly recommending this paper for acceptance. The response from the authors is helpful but doesn't raise my score to a 6.

**Limitations:**

Yes

**Quality:**

3

**Strengths And Weaknesses:**

This paper is clearly a step in the right direction and is far more transparent and statistically detailed than any other similar work I am aware of. The paper is relatively narrowly scoped to one model and one framework, but that's likely a de facto necessity because a more detailed analysis would be a mind-boggling amount of work. While this level of rigor would cause me to be critical in other areas, I don't fault the authors for balancing resource availability and detail, and praise them for raising the bar in this subfield.

The paper writing is pretty clean. On a conceptual level the work isn't very original, but that's a non-issue for a paper focused on providing systematic evidence of things that are vaguely known and often indirectly alluded to.

---

> ### Author Rebuttal · Authors · 2025-07-31
>
> We sincerely thank Reviewer DhYQ for the thoughtful and constructive feedback, and for recognizing the transparency and statistical depth of our work. We are encouraged by the positive assessment regarding our reproducibility focus and overall impact on the LLM web agent training subfield.
>
> ## General comments
> Since the submission deadline, we have significantly expanded and strengthened our results:
> * **Expanded the hyperparameter search for the bootstrap experiment (from 240 to 1,370 runs):** We increased the size of the random hyperparameter sweep by more than 5x. Importantly, the main conclusions reported in the original submission remain unchanged, demonstrating the robustness of our findings. We believe these results showcase that the proposed bootstrap technique is robust and sample-efficient, effectively finding the same conclusions even when scaling the number of runs 5x. The only minor update is that using standard-deviation normalized advantage appears moderately better than not using it when starting RL from the instruct model.
>
>
> * **Generality across benchmarks:** We performed additional WorkArena experiments. Our updated results confirm that the main conclusion from MiniWoB++—that combining SFT and RL achieves the best absolute performance—also holds on WorkArena. We could also get some stable runs for Pure RL, improving the held-out task performance by 7.5%.
>
> * **Generality across model families:** We ran experiments on Qwen2.5 and replicated the main conclusion: SFT+RL is the only method able to match or exceed teacher performance. This provides evidence that our findings generalize beyond the LLaMA family of models.
>
> * **Bootstrap reliability:** We discuss the consistency and unbiasedness of the bootstrap estimator as well as the statistic we are specifically interested in estimating. Additionally, we provide examples of the reliability of our estimate in terms of standard error as the number of runs increases, showing that even under pessimistic assumptions we can maintain a relatively small standard error.
>
> ## Reviewer-specific comments
>
> *On a conceptual level the work isn't very original, but that's a non-issue for a paper focused on providing systematic evidence of things that are vaguely known and often indirectly alluded to.*
>
> We appreciate this observation and would welcome suggestions from the reviewer on specific directions or techniques that could further improve the paper's originality. Our intention was to raise the bar of empirical rigor in this subfield, and we believe that systematically validating "vaguely known" insights at scale is itself a valuable contribution.
>
> *Some other works I've read seem to present mutually contradictory accounts of how important SFT before RLHF is. The general vibe I've gotten is that it may depend on the base model. Do you have any thoughts on this? Do you have intuition for if you'll see similar results if you use DeepSeek or Qwen?*
>
> Given our new results, we find that SFT before RL is generally beneficial for both Qwen and LLaMA models. The contradictory accounts in prior work may stem from the fact that too much SFT can hinder learning in the RL stage, leading to diminishing returns. We observe this same phenomenon: after a certain amount of SFT, additional SFT provides less benefit.
> To further address generality, we ran experiments with the Qwen2.5 model family. We performed a very restricted hyperparameter random search on MiniWoB++, only sweeping over learning rate, batch size, and sampling temperature (all other hyperparameters were fixed to those identified as optimal in the LLaMA bootstrap). The updated MiniWoB++ results are shown below:
>
> | Model                           | Heldout Goals     | Heldout Tasks     |
> |---------------------------------|-------------------|-------------------|
> |---------------------------------|-------------------|-------------------|
> | Llama-3.3-70B Instruct (teacher)| 63.2 ± 2.4        | 61.9 ± 2.4        |
> | Llama-3.1-8B Instruct           | 29.5 ± 2.3        | 36.4 ± 2.4        |
> | Llama-3.1-8B SFT                | 53.4 ± 2.5        | 55.6 ± 2.5        |
> | Llama-3.1-8B RL                 | 43.5 ± 2.5        | 43.5 ± 2.5        |
> | Llama-3.1-8B SFT+RL             | **66.3 ± 2.4**    | **62.9 ± 2.4**    |
> |---------------------------------|-------------------|-------------------|
> | Qwen-2.5-72B Instruct (teacher) | 61.0 ± 3.4        | 59.0 ± 3.4        |
> | Qwen-2.5-7B Instruct            | 32.8 ± 3.3        | 37.0 ± 3.4        |
> | Qwen-2.5-7B SFT                 | 57.0 ± 3.5        | 56.5 ± 3.5        |
> | Qwen-2.5-7B RL                  | 32.8 ± 3.3        | 37.0 ± 3.4        |
> | Qwen-2.5-7B SFT+RL              | **63.5 ± 3.4**    | **62.0 ± 3.4**    |
>
>
> We were able to replicate the MiniWoB++ result that SFT+RL achieves the best absolute performance with a new model family, namely Qwen2.5. It is also the only method able to surpass the teacher. Interestingly, we could not improve the performance of the instruct model using Pure RL in this case. After a few steps, the model often failed to follow the correct output format, began producing repeated tokens, or switched to unrelated languages. Nevertheless, the conclusions drawn from the LLaMA experiment still hold.
>
> *A key issue for deployment is the robustness of a model across a wide diversity of website structures. Arguably a 1% failure rate for a general purpose web agent is problematically high. Have you considered this or do you plan to assess this?*
>
> We agree this is a critical point. While human error rates can be higher for these tasks, autonomous systems are typically held to a higher standard. At present, no models achieve a 100% success rate on any of the reported benchmarks. As models improve, it is essential for the community to have access to high-performing open models that enhance data privacy and reduce costs, which is the main motivation of our work.
> We also believe that it is important for models to be able to detect when they cannot complete a task and defer to a human or another system for assistance. This capability remains an open challenge for modern LLMs, and we believe that progress on this front will directly benefit web-based agents as well.

---

> > ### Comment · Reviewer_DhYQ · 2025-08-08
> > **Thank you**
> >
> > Thank you for the response. The bar for a 6 is very high, so I will be leaving my rating at a 5. This is a very good paper and I look forward to advocating for its acceptance should that be necessary.

---

### Public Comment · ~Massimo_Caccia1 · 2025-10-29
**Forgot to add the Qwen compute plots in the camera ready. Added on the Arxiv version**

Hello,

Thank you again for updating your scores based on our rebuttal results. We realized we forgot to include the Qwen compute plot in the camera-ready version. Our apologies for the oversight. It is included in the Arvix version now https://arxiv.org/pdf/2507.04103

Sorry again for the omission.

---

### Note · Authors · 2025-08-15

We thank the AC and reviewers for their constructive engagement, which has significantly strengthened our work.
## Summary of new contributions since the rebuttal:
Following the rebuttal, we extended our study to Qwen 2.5, running the full compute allocation analysis previously performed for LLaMA 3.1. These experiments confirm that **SFT + RL remains Pareto-optimal** for this new model family, demonstrating the **generality of our compute-allocation findings across models**. We are now expanding our bootstrap-based random search analysis for both LLaMA 3.1 and Qwen 2.5, and will report these results, along with updated compute allocation plots, in the camera-ready.
| Compute (total ×10¹⁸ FLOPs) | Train | Test |
|-|-|-|
| Instruct (0)                | 0.37  | 0.40 |
|                             |       |      |
| Pure RL (0.34)              | 0.50  | 0.53 |
| Pure SFT (2.61)             | 0.42  | 0.49 |
|                             |       |      |
| SFT 2.61 + RL (4.06)        | 0.57  | 0.57 |
| Pure SFT (5.23)             | 0.46  | 0.46 |
|                             |       |      |
| SFT 5.23 + RL (7.00)        | 0.57  | 0.58 |
| SFT (7.85)                  | 0.57  | 0.56 |
|                             |       |      |
| SFT 7.85 + RL (9.90)        | 0.64  | 0.61 |
| SFT (13.1)                  | 0.62  | 0.61 |

Results are averaged over two seeds.
## Per-reviewer synthesis:
* R1 (DhYQ) – Praised rigor, transparency, and statistical depth; requested another model family—addressed with strong Qwen 2.5 results. Score unchanged but willing to advocate.
* R2 (8JSX) – Initially questioned the generality of our findings. Our rebuttal provided new results validating our methodology across both models and benchmarks, prompting a score increase. They suggested adding a new branching-timing experiment with a different model, which we are now including in this Final Remark.
* R3 (zHWC) – Raised generalizability concerns; new Qwen 2.5 results, expanded sweeps, and additional analysis addressed these, leading to a higher score.
* R4 (jEdW) – Sought robustness evidence, broader sweeps, cross-model validation; we expanded sweeps from 240 → 1,370 runs (no change in conclusions), validated bootstrap, and generalized to Qwen 2.5 and WorkArena. No post-rebuttal engagement, but concerns addressed.
* R5 (Kh9s) – Valued compute allocation curve and statistical analysis; post-rebuttal clarifications, expanded sweeps, and Qwen 2.5 validation increased score.

---

### Decision · Program_Chairs · 2025-09-17

**Decision:**

Accept (poster)

**Comment:**

This paper makes a valuable empirical contribution to training LLM web agents by providing the first systematic analysis of compute allocation between supervised fine-tuning and reinforcement learning. The authors used extensive experiments to study resource allocation for SFT and RL for training LLM web agents. After rebuttal, most reviewers agree that this paper makes significant contributions to the community. Therefore, I recommend acceptance.